# *Pro3D-Editor*: A Progressive-Views Perspective for Consistent and Precise 3D Editing

**Yang Zheng[1]  Mengqi Huang[1]* Nan Chen[1]  Zhendong Mao[1,2]**
[1]University of Science and Technology of China
[2]Institute of Artificial intelligence, Hefei Comprehensive National Science Center
{zy849900389,huangmq,chen_nan}@mail.ustc.edu.cn, zdmao@ustc.edu.cn

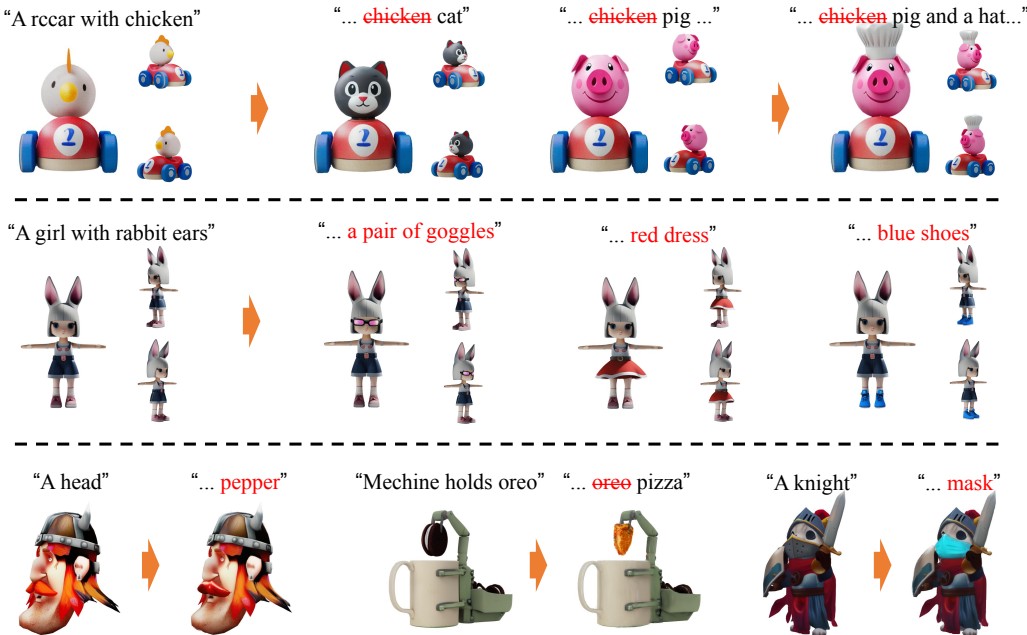

Figure 1: We propose a new 3D editing pipeline, Pro3D-Editor, which achieves consistent and precise local editing of 3D objects: our method supports iterative editing of 3D objects (top), handles diverse local edits on the same 3D object effectively (mid), and demonstrates strong editing capabilities across various 3D objects (bottom).

## Abstract

Text-guided 3D editing aims to precisely edit semantically relevant local 3D regions, which has significant potential for various practical applications ranging from 3D games to film production. Existing methods typically follow a view-indiscriminate paradigm: editing 2D views indiscriminately and projecting them back into 3D space. However, they overlook the different cross-view interdependencies, resulting in inconsistent multi-view editing. In this study, we argue that ideal consistent 3D editing can be achieved through a *progressive-views paradigm*, which propagates editing semantics from the editing-salient view to other editing-sparse views. Specifically, we propose *Pro3D-Editor*, a novel framework, which

*Mengqi Huang is the corresponding author

39th Conference on Neural Information Processing Systems (NeurIPS 2025).

mainly includes Primary-view Sampler, Key-view Render, and Full-view Refiner. Primary-view Sampler dynamically samples and edits the most editing-salient view as the primary view. Key-view Render accurately propagates editing semantics from the primary view to other key views through its Mixture-of-View-Experts Low-Rank Adaption (MoVE-LoRA). Full-view Refiner edits and refines the 3D object based on the edited multi-views. Extensive experiments demonstrate that our method outperforms existing methods in editing accuracy and spatial consistency. Project Page: https://shuoyueli4519.github.io/Pro3D-Editor.

# 1  Introduction

*Text-guided 3D editing* [1, 2, 3, 4, 5, 6] aims to precisely edit specific local features of a given 3D object based on the text guidance while preserving all other text-irrelevant features. Recently, this task has attracted significant attention as it facilitates diverse and personalized 3D asset synthesis, bringing various practical applications ranging from 3D games to film production. Unlike the well-studied 2D editing [7, 8, 9, 10], text-guided 3D editing presents greater challenges as it demands a comprehensive understanding of real-world 3D structures to achieve both *inter-view consistency* (*i.e.*, ensuring coherent appearance across views) and *intra-view discrimination* (*i.e.*, enabling distinctive and view-specific edits for each view).

Existing methods focus on lifting editing semantics from the 2D image plane to the 3D spatial space, which can be categorized into two streams, *i.e.*, the *iterative single-view* stream and the *parallel multi-views* stream. The former stream [1, 11, 2] iteratively refines the 3D representation by leveraging gradients from individual view images until the 3D object is well-aligned with the textual guidance, as shown in Fig. 2 (a). For example, Vox-E [11] uses a pre-trained text-to-image diffusion model to obtain each view's gradients and then repeatedly update the 3D object. The latter stream [3, 4, 5, 6] simultaneously edits multiple rendered images from fixed viewpoints and subsequently propagates these modifications onto the 3D object, as illustrated in Fig. 2(b). For example, PrEditor3D [5] employs prompt-to-prompt image editing to modify rendered multi-view images from fixed viewpoints, and then update the 3D objects. In summary, the commonality of both streams is that they are view-indiscriminate, *i.e.*, each view of the 3D object is edited indiscriminately.

However, the existing view-indiscriminate paradigm overlooks the different **cross-view interdependencies** induced by different editing instructions and therefore leads to view-conflicts, resulting in inconsistent 3D editing. Naturally, each view of the 3D object shows different editing salience depending on the editing instruction. For instance, "adding glasses" to a 3D character primarily affects its frontal view with minimal impact on its rear one, while "adding a ponytail" conversely. Therefore, the cross-view interdependence manifests as the **editing interdependence across views**, where an "editing-salient" view is more effective in guiding an "editing-sparse" one, while the reverse can only provide insufficient guidance and therefore lead to view-conflicts. As shown in Fig. 2(d), the existing *iterative single-view* stream indiscriminately samples a random view to edit at each step, disregarding its editing salience, result in view-conflicts where both the front and back views erroneously display a cat face (highlighted by the red bounding box). Meanwhile, as shown in Fig. 2(e), the existing *parallel multi-view* stream indiscriminately samples several fixed views and edits each indiscriminately, ignoring the varying semantic salience of different views with respect to the editing instruction, thereby leading to conflicts among these edited views, such as a pizza appearing in the frontal view but disappearing its rear one (highlighted by the red bounding box).

To address these challenges, we propose a novel **progressive-views paradigm**, which progressively samples and edits views from editing salient to sparse, enabling a consistent and smooth editing process for arbitrary 3D objects and editing instructions. Compared with the *iterative single-view* stream, our paradigm edits views in descending order of salience, avoiding the conflicts caused by random view sampling (highlighted by the blue bounding box in Fig. 2(d)). Compared with the *parallel multi-view* stream, our paradigm first edits the salient views and then uses them to guide further sparse view editing, avoiding the conflicts caused by indiscriminately editing multiple views in parallel (highlighted by the blue bounding box in Fig. 2 (e)).

Technically, we propose a novel **pro**gressive **3D** editing framework termed **Pro3D-Editor**, which constructs a hierarchical "primary-view → key-views → full-views" editing pipeline based on the dynamic editing salience across different views. Specifically, the Pro3D-Editor consists of three

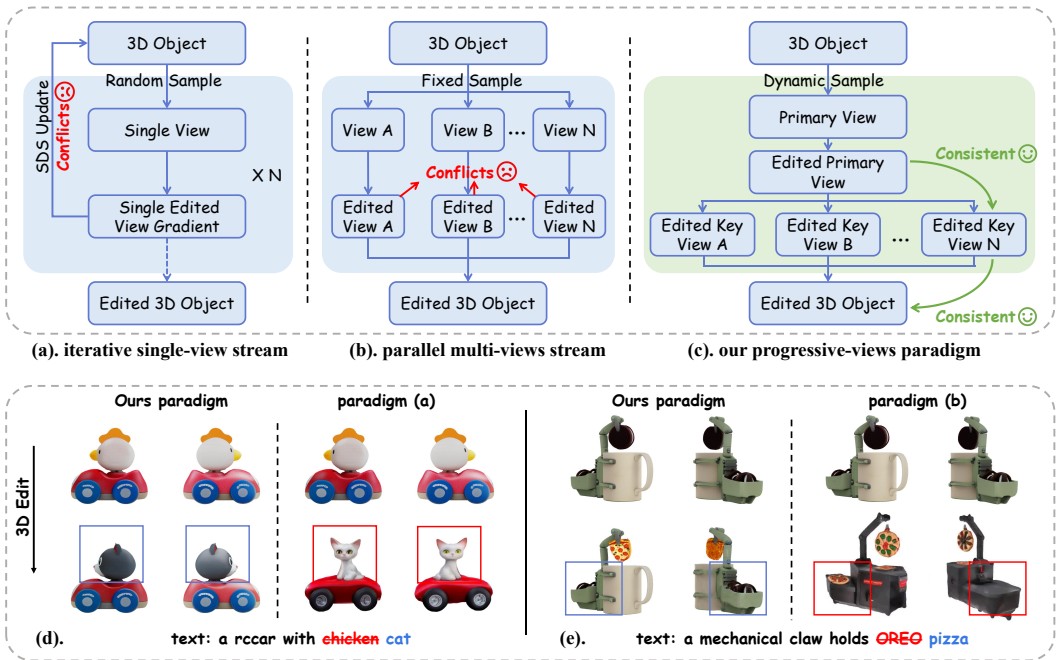

Figure 2: We propose a novel editing paradigm (top) for text-guided 3D editing. Compared with existing paradigms, it achieves spatial consistency in edited regions (d) and mitigates feature conflicts across views (e). Moreover, our paradigm enables more precise local 3D editing.

successive modules: (1) *Primary-view Sampler* module dynamically samples and edits the most editing salient view as the primary view by calculating the salience score between each view and the editing signal, which is further linearly extrapolated with its corresponding negative view to amplify accuracy. (2) *Key-view Render* module takes the edited primary view as the anchor and propagates its editing semantics to other key views. This is achieved through a novel Mixture-of-View-Experts Low-Rank Adaption (MoVE-LoRA), which learns feature correspondences from the primary view to the remaining key views while blocking reverse learning to avoid conflicts. (3) *Full-view Refiner* module repairs numerous newly rendered views to refine the edited 3D result, which is achieved by fusing the editing information from the edited key multi-views.

Our main contributions are summarized as follows: (1) **Concepts.** We introduce a *progressive-views paradigm* for consistent and precise 3D editing by propagating the editing semantics from editing-salient views onto editing-sparse views. (2) **Technology.** Based on the proposed paradigm, we design a pipeline called *Pro3D-Editor*. In this pipeline, the Primary-view Sampler dynamically samples the most editing-salient view by calculating salience scores. The Key-view Render captures feature correspondences from the editing-salient view to the editing-sparse views while blocking reverse learning to avoid conflicts. The Full-view Refiner repairs numerous newly rendered views to provide additional 3D structural information, refining the edited 3D regions. (3) **Experiments.** Extensive experimental results demonstrate that *Pro3D-Editor* surpasses current methods, achieving a 47.4% improvement in LPIPS (editing quality) and a 9.7% improvement in DINO-I (editing accuracy).

## 2    Related Works

**Multi-View Generation Models** generate multi-view images guided by a single 2D input. Trained on large-scale 3D datasets [12, 13], multi-view diffusion models [14, 15, 16, 17, 18, 19, 20, 21] effectively capture the spatial relationships across multi-views. Zero-1-to-3 [22] first encodes external camera parameters to generate multi-view images from specified perspectives. MVDream [23] introduces multi-view attention mechanism to extend self-attention mechanism to 3D, improving spatial consistency across multi-views. Building on these, we propose fine-tuning models to better align features between the primary and other views, enabling precise and consistent regional multi-view editing.

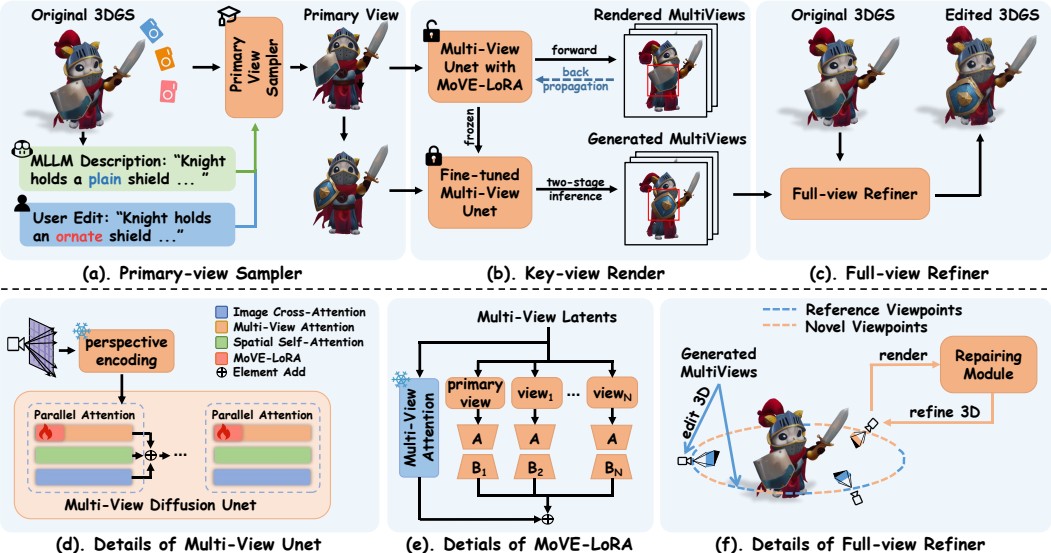

Figure 3: **Method overview**. Given a 3D object represented by 3DGS, *Pro3D-Editor* achieves precise 3D editing, which includes three main steps: (a) Primary-view Sampler selects and edits the most editing-salient view as the primary view. (Sec. 3.1); (b) Key-view Render accurately propagates the editing information from the primary view to local regions of the remaining key views. (Sec. 3.2); (c) Full-view Refiner edits and refines the 3D object based on the edited multi-views. (Sec. 3.3).

**3D Reconstruction From Multi-Views** aims to generate 3D objects from given images, which can be naturally extended to the 3D editing task. Mainstream 3D representations include NeRF, triplane, and 3D Gaussian Splatting (3DGS). NeRF [24, 25] encodes 3D scenes implicitly with MLPs trained on dense views. Triplane models [26, 27, 28, 29, 30, 31] represent features on orthogonal planes. 3DGS [32, 33] explicitly models 3D objects as collections of Gaussians, iteratively refined with multi-view supervision. In 3D editing, it is crucial to modify only the edited regions. Implicit methods like NeRF and triplane struggle with this, whereas the iterative nature of 3DGS makes it especially suitable. Therefore, we use 3DGS as our editing representation.

## 3 Method

The pipeline of *Pro3D-Editor* is shown in Fig. 3, including three main modules: Primary-view Sampler, Key-view Render, and Full-view Refiner. Primary-view Sampler is designed to sample and edit the primary view for subsequent multi-view editing (Sec. 3.1). Key-view Render is designed to accurately propagate the editing semantics from the primary view to other key views, achieving precise multi-view editing (Sec. 3.2). Full-view Refiner repairs numerous newly rendered views by fusing the editing information from the key views, which helps address the fragmentation issue in sparse-views guided 3DGS editing and achieve high-quality 3D editing(Sec. 3.3).

### 3.1 Primary-view Sampler

Primary-view Sampler selects the most editing-salient view (*i.e.*, the one with the richest editing information) as the primary view and edits it. When editing multi-views, the fine-tuned multi-view diffusion model propagates the editing information from the primary view to the remaining key views. Therefore, the choice of the primary view significantly affects the quality of the final editing results.

As shown in Fig. 3 (a), the 3D object is first rendered into a continuous 360° surrounding video, denoted as $\boldsymbol{I_c} \in \mathbb{R}^{F \times H \times W}$, where $F$ represents the number of rendered images, $W$ and $H$ represent the width and height of rendered images. The frames are then fed into the Multimodal Large Language Model [34] to obtain descriptive text $y_s$ for the original 3D object. And the user-provided editing text is denoted as $y_e$. Our Primary-view Sampler evaluates each rendered image by considering the relationships among $y_s$, $y_e$, and $\boldsymbol{I_c}$, assigning a score to each image. The rendered image with the highest score is selected as the primary view for the entire 3D editing pipeline.

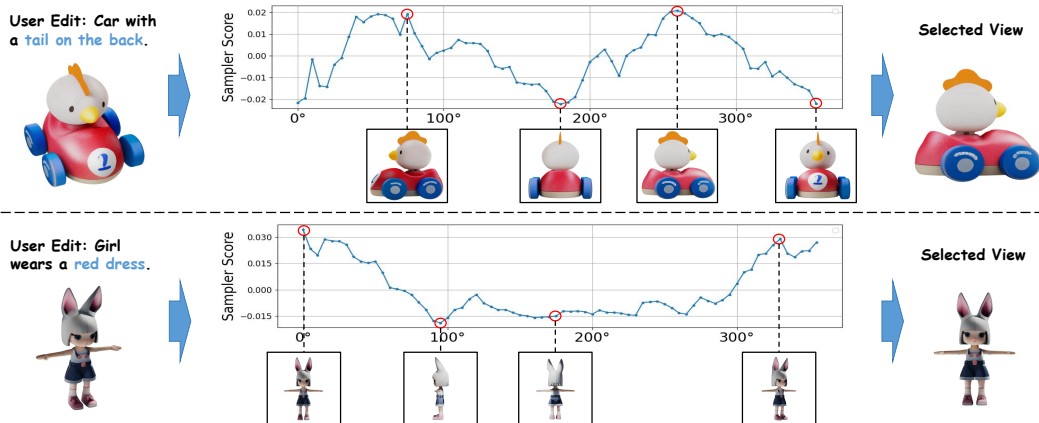

Figure 4: **Score distribution** of Primary-view Sampler. It automatically selects the most editing-salient view as the primary view based on the given 3DGS object and user-provided editing prompt. When editing a tail, the scores exhibit a reasonable bimodal distribution, peaking at the side views. When editing the dress, the scores show a reasonable unimodal distribution, peaking at the front view.

Primary-view Sampler focuses on two aspects. First, the selected view should align well with the text, ensuring high editing information density. Propagating edits from editing-salient views to editing-sparse views helps reduce feature conflicts across views. Second, the relative views at 135° and 225° azimuth angles with respect to the primary view should contain minimal information, serving as a penalty term. This is because the fine-tuning data for the multi-view generation model includes six fixed-perspective rendered images, with the azimuth angle of the first view set at 0°, and the remaining five views at 45°, 90°, 135°, 180°, and 225°, respectively. The lack of views at 135° and 225° means less information on the back side during 3D editing. If these missing views contain significant editing information, it can greatly reduce the quality of the final 3D edit.

Given the above considerations, the scoring formula of Primary-view Sampler can be expressed as:

$$\text{score}^i \leftarrow P(\boldsymbol{I_c}, y_s)^i + P(\boldsymbol{I_c}, y_e)^i - \alpha \times (P(\boldsymbol{I_c}, y_s, y_e)^p + P(\boldsymbol{I_c}, y_s, y_e)^q), \tag{1}$$

where $P(\boldsymbol{I_c}, y_s)^i = \text{softmax}(\text{CLIP}(\boldsymbol{I_c}, y_s))^i$, softmax transforms these CLIP similarities into probabilities for selection and $i$ represents the i-th view. $P(\boldsymbol{I_c}, y_e)^i = \text{softmax}(\text{CLIP}(\boldsymbol{I_c}, y_e))^i$. $P(\boldsymbol{I_c}, y_s, y_e)^p$ = $P(\boldsymbol{I_c}, y_e)^p$ - $P(\boldsymbol{I_c}, y_s)^p$. $p$ and $q$ represent the views at relative angles of 135° and 225° with respect to the i-th view. $\alpha$ is a hyperparameter that controls the weight allocation.

Primary-view Sampler selects the highest-scoring rendered image as the primary view, denoted as $\boldsymbol{c_i} \in \mathbb{R}^{1 \times H \times W}$. Subsequently, a 2D editing model edits this primary view based on the provided editing text, generating the edited image $\boldsymbol{c_e}$, which serves as the edited primary view for subsequent multi-view editing. As shown in Fig. 4, the Primary-view Sampler is capable of selecting the most editing-salient view and exhibits reasonably low scores on editing-sparse views.

## 3.2 Key-view Render

Under the guidance of the edited primary view $\boldsymbol{c_e}$ obtained from Sec. 3.1, Key-view Render accurately propagates editing information from the primary view to local regions of the remaining key views. To ensure accurate editing region control and feature consistency in edited regions, we introduce improvements to both fine-tuning and inference stages of the multi-view diffusion model. In the fine-tuning stage, we design a Mixture-of-View-Experts Low-Rank Adaption (MoVE-LoRA) to capture feature correspondences from the primary view to the remaining key views, which accurately edits the semantically consistent local regions of the remaining key views based on the editing information from the primary view. In the inference stage, we adopt a two-stage inference strategy to further enhance feature consistency within edited regions.

### 3.2.1 Mixture-of-View-Experts Low-Rank Adaption

Our Mixture-of-View-Experts Low-Rank Adaption (MoVE-LoRA) is introduced as an additional trainable component to fine-tune the backbone. It is designed to capture feature correspondences from the primary view to the remaining views, laying the foundation for accurate multi-view editing.

As shown in Fig. 3 (b), our multi-view generation backbone [35] contains a parallel attention module with three components: image cross-attention, multi-view attention, and spatial self-attention. Among these attention components, only the multi-view attention focuses on the feature correspondence between multi-view images. To capture accurate corresponding features from the primary view to the remaining views, we utilize the LoRA [36] structure exclusively within the multi-view attention. However, since each view exhibits distinct feature correspondences with the primary view, sharing the same LoRA weights across multi-view images can lead to feature entanglement, thereby hindering accurate feature alignment across views. To address this, we design MoVE-LoRA to decouple the feature correspondences among multi-views.

The detail of MoVE-LoRA is shown in Fig. 3 (d). A shared matrix $A \in \mathbb{R}^{r \times d}$ is designed to capture the features of the primary view, where $d$ denotes the number of channels in the image latent and $r$ denotes the low-rank dimension. Different matrices $B_i \in \mathbb{R}^{d \times r}$ represent different expert models, which are used to capture the distinct feature correspondences between each view and the primary view, decoupling the features among multi-views. The forward process can be expressed as:

$$y = W_0 x + \sum_{i=1}^{M} B_i A x_i, \tag{2}$$

where $W_0$ denotes the Linear layers in the attention module. $M$ denotes the number of experts, which is set to be equal to the number of multi-views. $x$ denotes the multi-view image latents.

Specifically, the matrix $A$ is updated via backpropagation solely from the primary view, without any gradient contributions from the remaining key views. This design encourages the matrix $A$ to learn the intrinsic features of the primary view and facilitates each matrix $B_i$ to capture the feature mappings from the primary view to the remaining key views.

In the multi-view diffusion model fine-tuning, the selected view $c_i$ from Primary-view Sampler and user-provided editing prompt $y_e$ serve as conditional inputs to the denoiser $\epsilon_\theta(\cdot)$. Then, with the viewpoint of the primary view set as the 0° azimuth, we render the remaining five key views. These six images are finally used as fine-tuning data to train our MoVE-LoRA, enabling it to capture feature correspondences from the primary view to the remaining key views. Our MoVE-LoRA structure is trained by mean-squared loss:

$$\mathcal{L} = \mathbb{E}_{z,\epsilon,t} \| \epsilon - \epsilon_\theta(z_t, t, y_e, c_i) \|_2^2, \tag{3}$$

where $\epsilon$ denotes unscaled noise and $t$ denotes denoising timestep.

### 3.2.2 Two-Stage Inference

While the fine-tuned multi-view diffusion model with MoVE-LoRA captures disentangled feature correspondences, it also learns redundant features in edited regions, compromising feature consistency across views. To mitigate this issue, we propose a two-stage inference approach, retaining the inherent spatial understanding capability of the backbone model when generating edited features.

The edited primary image $c_e$ guides the fine-tuned multi-view diffusion model to generate multi-view images. The editing regions in these generated views precisely correspond to those in the primary view. However, the redundant features may interfere with the generated views, resulting in spatially unreasonable edits. To address this, we first obtain multi-view editing masks $M_e \in \mathbb{R}^{6 \times H \times W}$ by comparing generated results with the original multi-view images. These masks represent the local editing regions that the multi-view diffusion model deems semantically relevant. Then, we incorporate these masks into the inference process to perform a second round generation, better preserving the backbone's inherent spatial understanding capability.

In the second round of generation, the multi-view editing masks are applied within the multi-view attention layer to conduct a fusion operation. This specific procedure is formulated as:

$$z = ((1 - \lambda) \times \mathrm{MA}(z) + \lambda \times \mathrm{MA}_{\mathrm{MoVE\text{-}LoRA}}(z)) \odot M_e + \mathrm{MA}_{\mathrm{MoVE\text{-}LoRA}}(z)) \odot (1 - M_e). \tag{4}$$

Table 1: **Quantitative comparison** with existing methods. *Pro3D-Editor* achieves the best performance in terms of both editing quality and precise manipulation of targeted 3D object regions.

| Methods | editing quality | | | | editing accuracy | |
|---|---|---|---|---|---|---|
| | FID ↓ | PSNR ↑ | LPIPS ↓ | FVD ↓ | CLIP-T ↑ | DINO-I ↑ |
| Tailor3D [37] | 139.53 | 12.39 | 0.323 | 1056.4 | 0.279 | 0.701 |
| MVEdit [3] | 176.64 | 13.99 | 0.362 | 1748.7 | 0.298 | 0.741 |
| 3D-Adapter [4] | 160.49 | 13.11 | 0.356 | 1236.7 | 0.292 | 0.726 |
| LGM [38] | 85.12 | 16.85 | 0.192 | 685.6 | 0.299 | 0.846 |
| Ours (naive) | 107.94 | 16.44 | 0.196 | 774.8 | 0.285 | 0.823 |
| Ours (Pro3D-Editor) | **63.99** | **22.01** | **0.101** | **449.9** | **0.304** | **0.928** |
| Improvement | Δ 24.8% | Δ 30.6% | Δ 47.4% | Δ 34.4% | Δ 1.7% | Δ 9.7% |

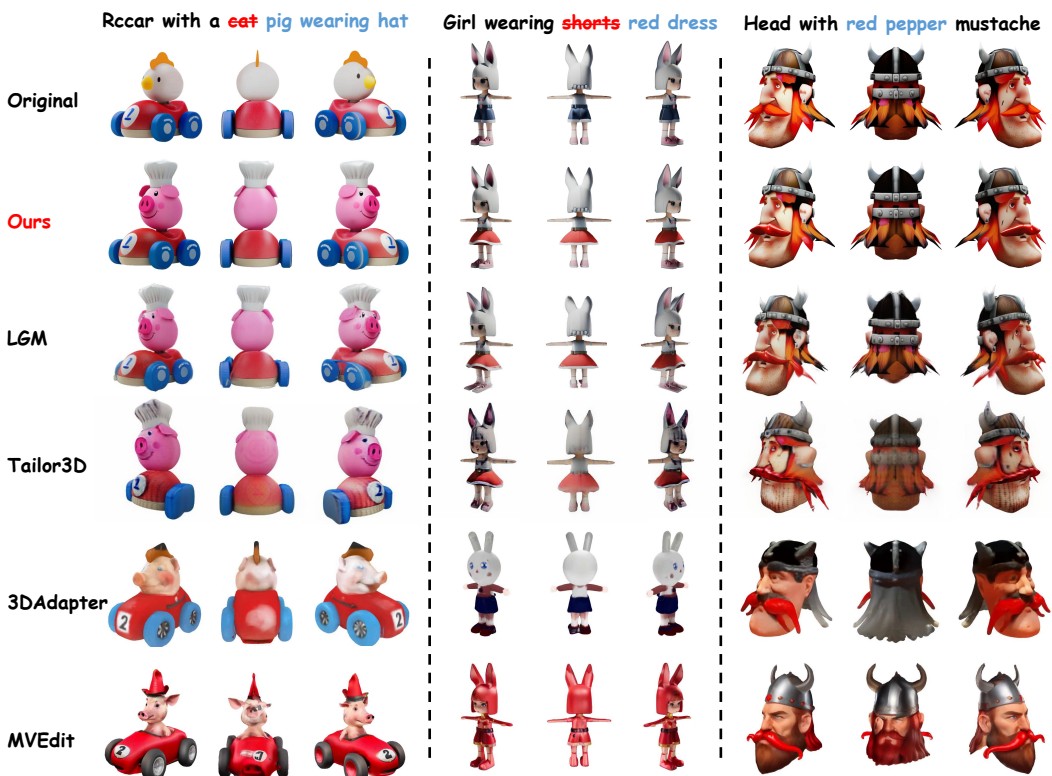

Figure 5: **Qualitative comparison** with existing methods. LGM and Tailor3D fail to preserve the original features, such as the shoes of a little girl doll. MVEdit inconsistently edits a new face on the back of the head. In comparison, *Pro3D-Editor* achieves accurate and spatially consistent 3D editing.

Here, $z \in \mathbb{R}^{6 \times h \times w}$ represents the multi-view latents, with $h$ and $w$ as height and width. MA denotes the multi-view attention layers in the backbone, while $MA_{\text{MoVE-LoRA}}$ denotes the fine-tuned versions of these layers. $M_e$ denotes the binary mask of the edited regions.

In summary, we establish accurate feature correspondences across multi-views through MoVE-LoRA and further enhance the feature consistency via a two-stage inference strategy. These edited key multi-view images serve as supervisory views for the subsequent Full-view Refiner.

### 3.3 Full-view Refiner

Under the guidance of the edited key images $I_e$ obtained from Sec. 3.2.2, our Full-view Refiner is designed to perform iterative editing of 3DGS objects and refine the edited 3D regions.

Similar to the past studies [39, 40] in the field of text-driven 3D scene editing, we directly project the edited key multi-views back into 3D space to achieve precise modifications in localized 3D regions. However, the discreteness and unstructured nature of 3DGS make it challenging to achieve high-quality editing solely based on sparse views, often leading to fragmented outcomes in edited 3D regions. Inspired by the field of sparse-views 3DGS reconstruction field [41], Full-view Refiner fuses the edited key multi-views to repair the newly rendered views, thereby obtaining more structured 3D information to help overcome the fragmentation issues in edited 3DGS.

First, we iteratively optimize the existing 3D object $\mathcal{O}$ under the guidance of $\boldsymbol{I_e}$ to obtain the preliminarily edited 3D object $\mathcal{O}_1$. While $\mathcal{O}_1$ preserves the original 3D features, it still exhibits fragmentation in the edited regions. Then, to repair fragmented editing regions, we utilize the multi-view images $\boldsymbol{I_r}$ rendered under $\mathcal{O}_1$ from the viewpoints corresponding to $\boldsymbol{I_e}$, and high-quality images $\boldsymbol{I_e}$ to fine-tune a repair module. The backbone of the repair module is a 2D diffusion model. During the training process, we add noise $\epsilon$ to $\boldsymbol{I_e}$ and get noised latents $\boldsymbol{z_t^e}$. The rendered degraded images $\boldsymbol{I_r}$ serve as the guidance condition to the denoising process. The loss function is defined as:

$$\mathcal{L}_{repair} = \mathop{\mathbb{E}}_{\boldsymbol{z},\boldsymbol{\epsilon},t} \|\boldsymbol{\epsilon} - \boldsymbol{\epsilon_\theta}(\boldsymbol{z_t^e}, t, y_t, \boldsymbol{I_r})\|_2^2, \tag{5}$$

where $y_t$ denotes an object-specific prompt, $t$ denotes denoising timestep.

The fine-tuned repairing module can learn how to generate structurally coherent views from degraded images. Finally, we render a large volume of images from novel viewpoints under the 3D object $\mathcal{O}_1$, which are then repaired by the fine-tuned model to provide additional 3D structural information for the edited regions. These numerous repaired novel views serve as the training data for iterative updates of $\mathcal{O}_1$, ultimately yielding a structured, high-quality 3D editing result $\mathcal{O}_2$.

## 4 Experiments

### 4.1 Experimental Setups

**Implementation Details.** The weighting coefficient $\alpha$ in the Primary-view Sampler is set to 0.5. For the MoVE-LoRA, the rank of the shared matrix $\boldsymbol{A}$ is set to 32. The number of expert matrices $\boldsymbol{B_i}$ is set to 6. The weighting coefficient $\lambda$ in the two-stage inference stage is set to 0.5. We employ a leave-one-out strategy, updating the 3DGS object using the edited multi-views by iteratively leaving out one view and training on the remaining views for 10k steps. Then we employ ControlNet-Tile as the base of the Full-view Refiner, injecting LoRA into all attention layers with rank = 64, and fine-tune it for 1800 steps with a learning rate of 1e-3. Finally, we continue updating the 3DGS object for an additional 10k steps. The entire editing process is trained on an A100 GPU for about 1.5 hours.

**Evaluation.** Our evaluation 3D dataset contains 6 objects and 15 editing prompts. To construct the evaluation image dataset, we render 72 views for each edited object by sampling azimuth angles every 5°. We evaluate our method and the baselines from two aspects: (1) Editing quality: FID [42], PSNR, LPIPS, FVD [43], and the texture details score from GPTEval3D [44]. (2) Editing accuracy: CLIP-T [45], DINO-I [46], the 3D plausibility score and the text-asset alignment score from GPTEval3D. Detailed explanations of the evaluation metrics are provided in Appendix A.

### 4.2 Main Results

In this section, we compare *Pro3D-Editor* with state-of-the-art methods (MVEdit [3], 3DAdapter [4], Tailor3D [37], and LGM [38]) using qualitative and quantitative analyses. For a fair comparison, the multi-views for Tailor3D and LGM are from our method. More details are provided in Appendix A.

**Quantitative results.** As shown in Tab. 1, our method outperforms existing baselines in both 3D editing quality and editing accuracy. Compared with existing methods, *Pro3D-Editor* achieves a 47.4% improvement in editing quality (LPIPS) and a 9.7% improvement in editing accuracy (DINO-I). Additionally, as shown in Fig. 6, our editing results are highly likely preferred by GPTEval3D in terms of text-asset alignment, 3D plausibility, and texture details.

**Qualitative results.** Fig. 5 shows a qualitative comparison between *Pro3D-Editor* and existing methods. Our method is capable of producing editing results with finer details. In contrast, baselines like Tailor3D exhibit inferior editing quality, such as the girl doll's dress. Our method also ensures

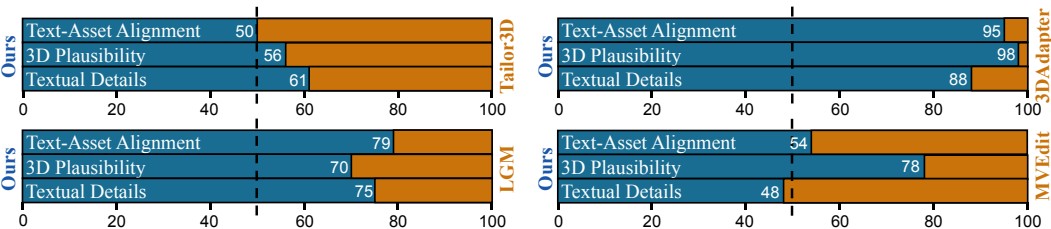

Figure 6: **Quantitative comparison** using GPTEval3D [44]. The blue segments indicate the selection rate of *Pro3D-Editor*, while the orange segments represent that of the baseline. A higher selection rate indicates better editing performance of the corresponding method.

Table 2: **Ablation Studies** of essential modules. Compared with the naive method (ID-0), introducing Primary-view Sampler (ID-1) enhances the alignment between the edited 3D objects and the editing prompts. Introducing MoVE-LoRA (ID-2) ensures spatial consistency in the edited regions. Full-view Refiner (ID-3) significantly improves the editing quality, with a 10.6% gain in the LPIPS metric.

| ID | Settings | LPIPS ↓ | CLIP-T ↑ | DINO-I ↑ |
|----|----------|---------|----------|----------|
| 0 | naive method | 0.196 | 0.285 | 0.823 |
| 1 | 0 + Primary-view Sampler | 0.118 | 0.300 | 0.875 |
| 2 | 1 + MoVE-LoRA | 0.113 | 0.302 | 0.879 |
| 3 | 2 + Full-view Refiner | **0.101** | **0.304** | **0.928** |

spatial consistency in the edited regions, while baselines like MVEdit often generate spatially inconsistent objects, such as facial features on the back of the head. Furthermore, our method accurately edits semantically relevant local regions, which the baselines fail to achieve.

## 4.3 Ablations

To evaluate the effectiveness of our proposed paradigm and the essential components in improving consistency and quality, we conduct extensive ablation experiments. It is important to note that the quantitative metrics used in these experiments are based on 2D image evaluations, which have certain limitations. Specifically, some spatial inconsistencies and structural discontinuities that are clearly noticeable in 3D space may lead to only minor numerical differences when projected into 2D space, making them difficult to detect through 2D metrics alone. Therefore, to provide a more comprehensive assessment, we provide additional visualizations in Appendix B.

**Effectiveness of progressive-views paradigm.** As shown in Fig. 5, we first explain our naive method, which uses fine-tuning for 3D editing. It replaces our Primary-view Sampler with a random sampling of the primary view, substitutes MoVE-LoRA with the simplest LoRA structure where all views share the same LoRA, and removes our Full-view Refiner. By comparing our *Pro3D-Editor* with this defined naive method, we demonstrate that it is our proposed progressive-views paradigm that ensures high-quality and accurate text-guided 3D editing, rather than the fine-tuning approach itself. Compared with the naive method, *Pro3D-Editor* achieves a 48.5% improvement in the LPIPS metric (editing quality) and an 12.6% improvement in the DINO-I metric (editing accuracy).

**Effectiveness of Primary-view Sampler.** As shown in Tab. 2, compared with the method with a random sampling of the primary view (ID-0), introducing the Primary-view Sampler achieves a 5.3% improvement in the CLIP-T metric, which demonstrates that Primary-view Sampler effectively enhances the alignment between the edited 3D objects and the editing prompts.

**Effectiveness of MoVE-LoRA.** As shown in Tab. 2, compared with the method without MoVE-LoRA (ID-0), introducing MoVE-LoRA (ID-1) enables precise control over the edited regions and achieves an improvement on the DINO-I metric. As shown in Fig. 7, we provide visual examples to better demonstrate the effectiveness of MoVE-LoRA. The front view serves as the primary view in this case. Without any fine-tuning, the back view lacks precise control over the edited regions. When shared LoRA employs the same matrices $A$ and $B$ for all views, it fails to preserve spatial consistency in the edited regions (*i.e.*, the ears are lengthened in the front view but appear shorter in the back view). In contrast, our proposed MoVE-LoRA achieves precise and spatial consistent local 3D editing.

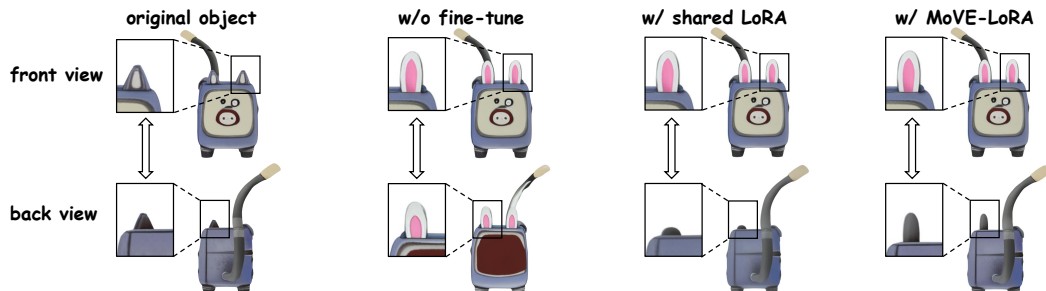

Figure 7: **Ablation Studies** of MoVE-LoRA. The edited multi-views generated with the MoVE-LoRA exhibit the most precise 3D local editing and superior spatial consistency in the edited regions.

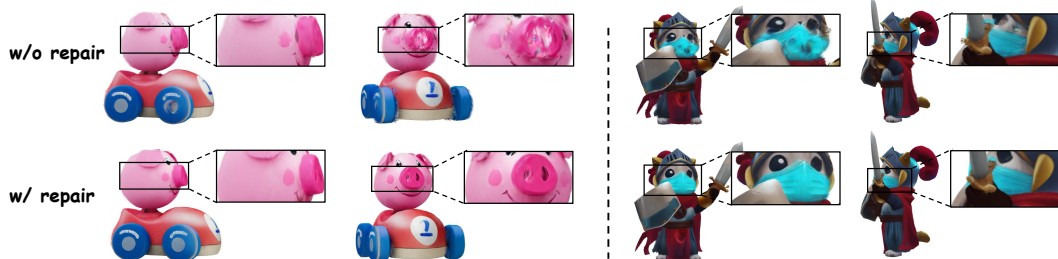

Figure 8: **Ablation Studies** of Full-view Refiner. Full-view Refiner mitigates structural fragmentation and blurriness, which are caused by directly applying sparse multi-view edits to existing 3D objects.

**Effectiveness of Full-view Refiner.** As shown in Tab. 2, compared with no Full-view Refiner (ID-2), introducing this module (ID-3) achieves a 10.6% improvement on the LPIPS metric, indicating enhanced perceptual quality. As shown in Fig. 8, when the edited multi-views are directly used to update the existing 3DGS objects, the edited 3DGS objects exhibit noticeable fragmentation and structural discontinuities (*e.g.*, the nose of a pig and the mask of a doll). In contrast, with Full-view Refiner, the edited 3D objects demonstrate greater structural continuity and improved detail.

## 5  Conclusion

In this paper, we propose a novel **progressive-views paradigm** to achieve consistent and precise text-guided 3D editing. Specifically, we design a corresponding pipeline **Pro3D-Editor**, which dynamically edits the most editing-salient view and propagates its editing semantics to other key views. Extensive experiments show our method outperforms the existing methods in spatial consistency and editing accuracy, demonstrating strong potential for 3D asset manipulation applications. More discussion about limitation and broader impacts are available in Appendix C.

# 6 Acknowledgements

This research is supported by Artificial Intelligence National Science and Technology Major Project 2023ZD0121200, and National Natural Science Foundation of China under Grant 623B2094 and 62222212.

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

# A Implementation Details and Comparative Experiments

## A.1 Implementation Details

We use the MV-Adapter SDXL checkpoint as our multi-view diffusion model. In our pipeline, we fine-tune the multi-view attention layers within the MV-Adapter network. For different views, we set distinct B matrices and identical A matrices, with the lora_rank set to 32 and lora_alpha set to 16. During training, the parameters of the A matrix are updated only by the gradients from the primary view. We fine-tune the model for 800 steps, which takes 45 minutes on an A100 GPU. During inference, we set the classifier-free guidance to 2. For 3D editing and refining, we first use a leave-one-out strategy to train the original 3DGS object for 10k steps, resulting in a degraded 3DGS. We then render the degraded views corresponding to the target perspectives and use them as the condition for ControlNet-Tile. Using the generated multi-views as the target, we add LoRA with a rank of 64 to all attention layers of the controlnet and fine-tune for 1800 steps. Finally, we use the fine-tuned ControlNet-Tile to repair the rendered images of new perspectives and train the degraded 3DGS for an additional 10k steps. The entire 3D editing and refining process takes about 45 minutes.

## A.2 Explanation of Quantitative Evaluation Metrics

In terms of evaluating editing quality, FID assesses the overall visual similarity between the edited result and the original object. LPIPS measures perceptual similarity, while PSNR reflects changes in detail. FVD evaluates the temporal continuity and stability across multi-views. The 3D plausibility score and texture details score proposed by GPTEval3D specifically measure the structural rationality and texture detail of 3D editing results. In terms of edit controllability, the text-asset alignment score from GPTEval3D and CLIP-T measure the similarity between the editing results and the editing text. DINO-I measures the similarity between the editing results and the original object. Since our task focuses on localized 3D editing, DINO-I can reflect the accuracy of the edits to some extent. Overall, these metrics provide a comprehensive quantitative evaluation of both the editing quality and editing accuracy from different perspectives, collectively reflecting the overall performance of the 3D editing method. However, when it comes to view consistency in the editing results, these metrics fall short of accurately reflecting it. Therefore, we provide additional visualizations to fully demonstrate the improvements of our method compared with existing baselines.

## A.3 Comparison with Existing Methods

In Fig. 9, we show a detailed editing example. Existing methods often edit the entire object and fail to preserve local regions that are semantically irrelevant to the editing text. Even though LGM and Tailor3D use multi-views generated from our method, they still modify semantically irrelevant regions. Moreover, existing methods such as MVEdit often generate spatially inconsistent 3D objects. In contrast, our method achieves consistent, precise, and high-quality text-guided 3D editing. For more comparison results, please refer to the HTML file provided in our supplementary materials, which contains multiple orbiting videos that demonstrate the improvements of our method in text-guided 3D editing.

# B More Ablation Experiments and User Studies

## B.1 Ablation of Each Component

**Effectiveness of Primary-view Sampler.** In Fig. 10 and Fig. 11, we highlight the importance of the Primary-view Sampler. When the primary view is randomly selected and editing semantics are propagated from an editing-sparse view to editing-salient views, it results in inter-view inconsistency (*i.e.*, lack of spatial coherence across views) and intra-view indiscrimination (*i.e.*, poor control over editing-salient views). These issues are clearly illustrated by the inconsistent beard appearance across views in Fig. 10 and the unreasonable editing of the cat's head in certain views in Fig. 11. It underscores the necessity of our progressive-views paradigm, which directs semantic flow from editing-salient to editing-sparse views. The precise and consistent 3D editing achieved by our method stems not merely from fine-tuning, but from this carefully designed paradigm.

**Effectiveness of MoVE-LoRA.** In Fig. 12, we present qualitative results to demonstrate the effec-

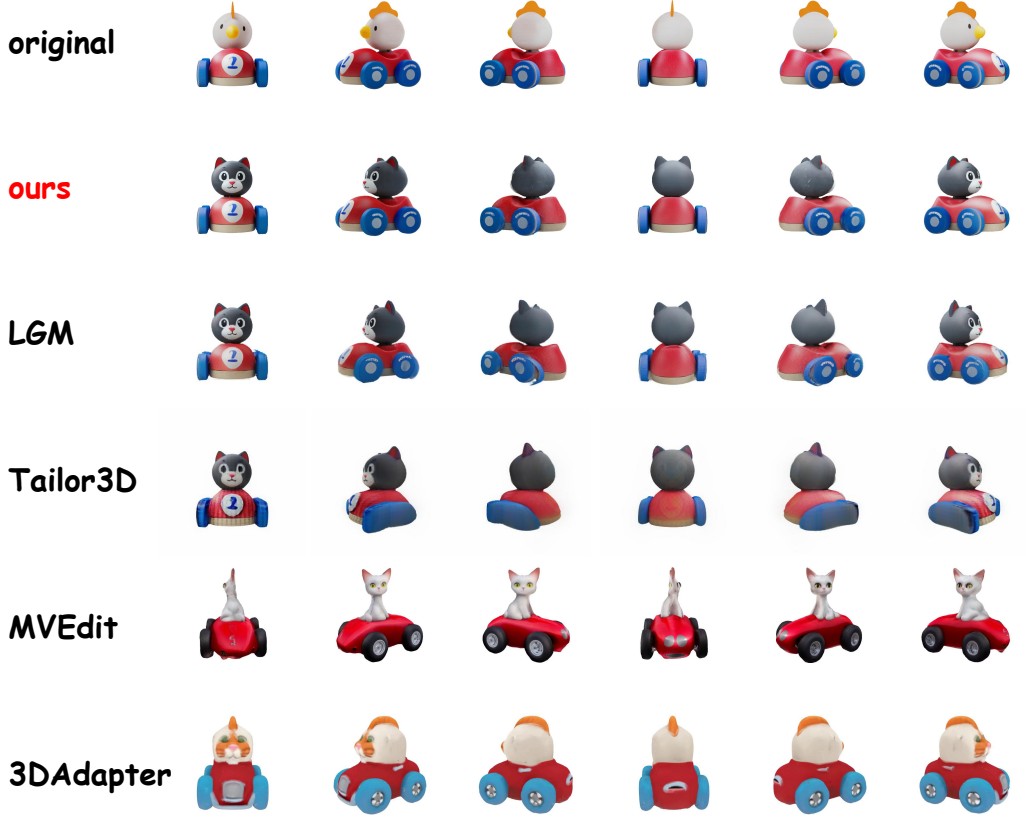

Figure 9: **Qualitative comparison** with existing methods. It can be observed that our method achieves precise and high-quality local 3D editing while addressing the issue of spatial inconsistency.

tiveness of MoVE-LoRA in enhancing multi-view editing consistency, as it is difficult to accurately evaluate such consistency using existing quantitative metrics. Here, "Shared LoRA" refers to a setting where the same LoRA matrices $A$ and $B$ are applied to the latent of multi-views. As shown in the figure, Shared LoRA fails to accurately preserve the original object features (e.g., incorrect object colors) and leads to spatially inconsistent edits (e.g., misaligned ears across views). In contrast, our MoVE-LoRA not only better preserves the original object features but also ensures spatially consistent editing across multi-views.

**Effectiveness of Full-view Refiner.** As shown in Fig. 13, we compare the 3D editing results with and without Full-view Refiner. Without Full-view Refiner, the edited 3DGS object may become fragmented. For example, in the case of a doll's mask, the absence of the Full-view Refiner can lead to the generation of numerous floating discrete Gaussians. This is because sparse-view guidance of 3DGS updates prioritizes consistency with the given multi-views at specific angles, potentially neglecting the overall 3D structural continuity. In contrast, introducing Full-view Refiner provides extra 3D structural information, ensuring the surface continuity of the final edited 3DGS object.

## B.2 Human Perception Evaluation

We recruit 8 volunteers to evaluate *Pro3D-Editor* under different settings from three aspects: Editing Consistency (EC), Editing Accuracy (EA), and Editing Quality (EQ). The volunteers were asked to rank the editing results under different settings from first to fourth place. Each volunteer is given two different edited objects to assess. As shown in Tab. 3, it can be observed that with the addition of each essential module, the final editing results align more closely with human preferences. Notably, the results in the table represent the average rankings given by all volunteers.

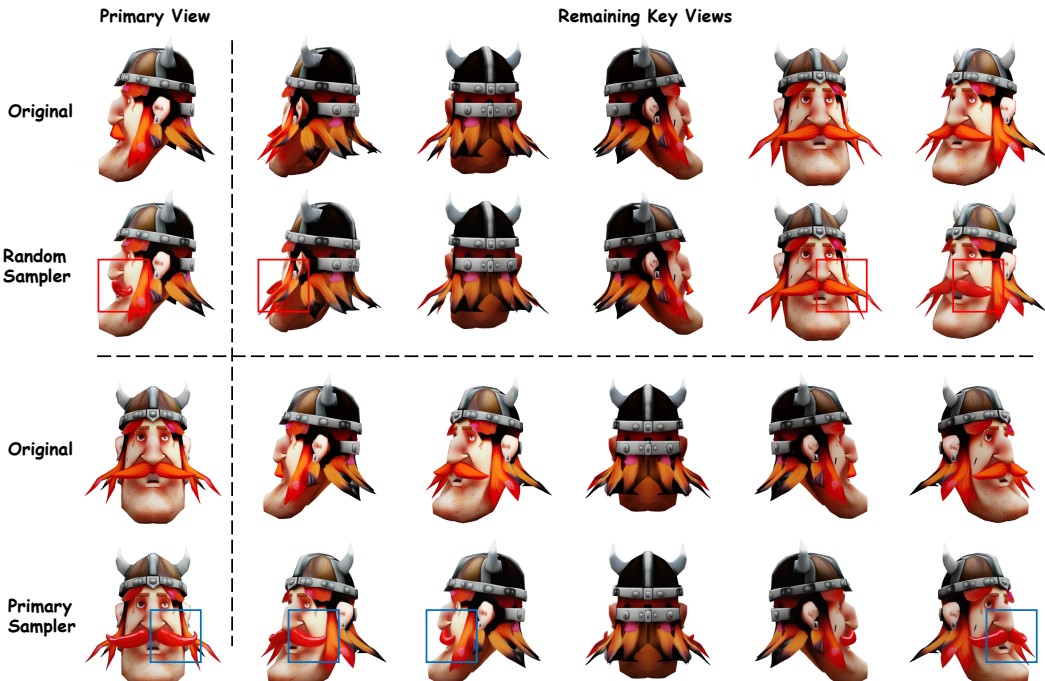

Figure 10: **Ablation studies** of Primary-view Sampler. When the randomly selected view is not the most editing-salient view, the editing information from this editing-sparse view may fail to propagate effectively to the editing-salient views, leading to spatially inconsistent across multi-views.

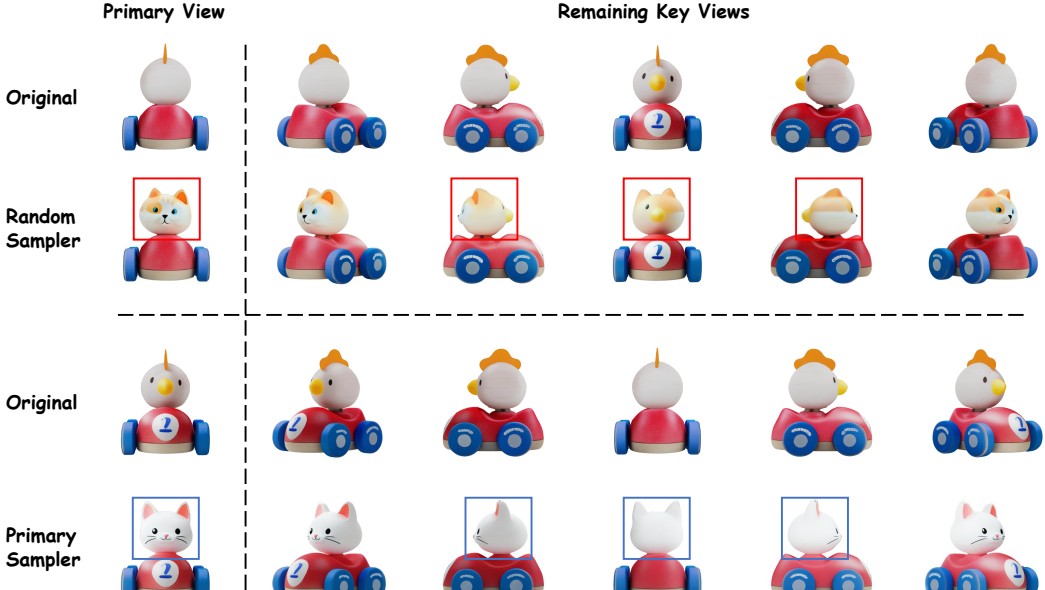

Figure 11: **Ablation studies** of Primary-view Sampler. Since editing-salient views are difficult to be precisely controlled by editing-sparse views, when the randomly selected view is not the most editing-salient view, the other editing-salient views may produce unreasonable editing results.

## C   Limitations and Broader Impacts

### C.1   Limitations

The *Pro3D-Editor* is computationally demanding and requires substantial GPU memory, primarily due to the fine-tuning process on a high-resolution multi-view generation model. Compared to

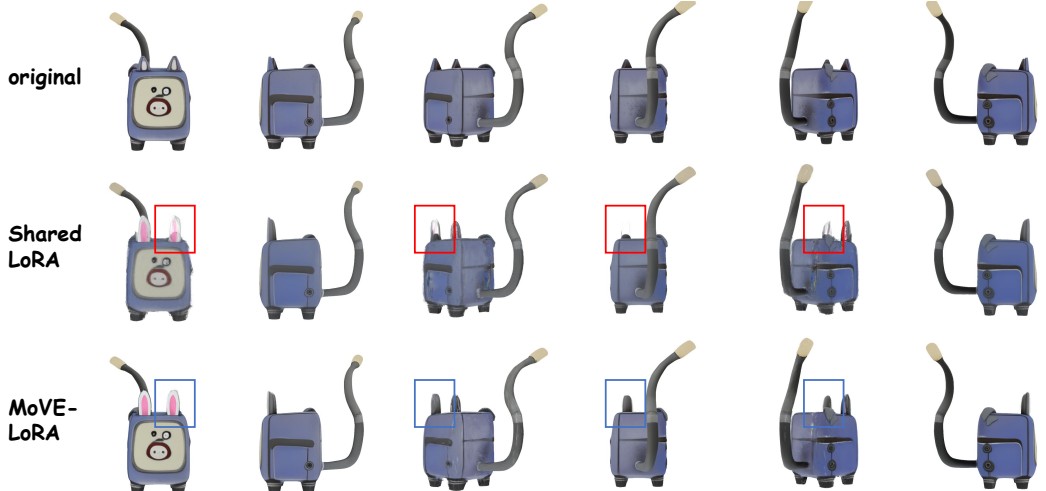

Figure 12: **Ablation studies** of MoVE-LoRA. Compared with Shared LoRA, our MoVE-LoRA not only better preserves the features of the original multi-views, but also ensures spatial consistency of the editing regions, achieving precise and consistent multi-view editing.

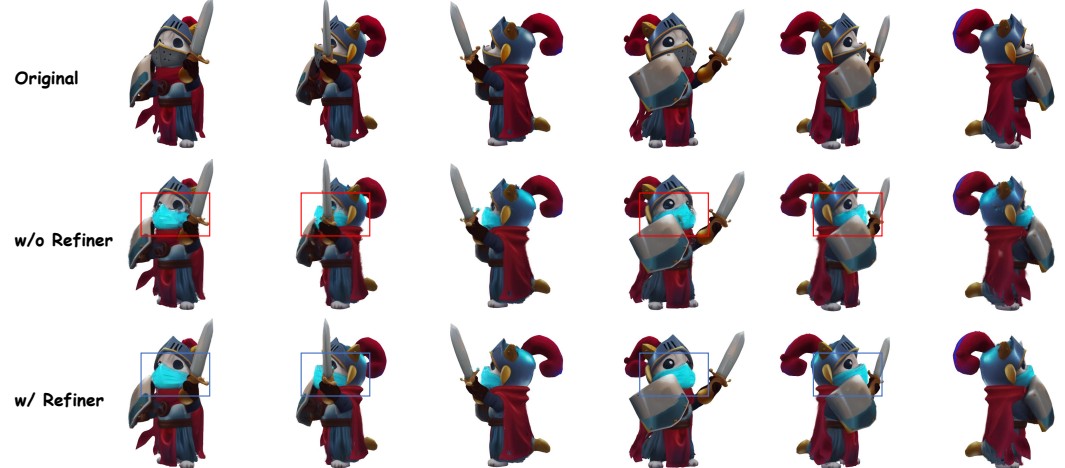

Figure 13: **Ablation studies** of Full-view Refiner. Introducing the Full-view Refiner can improve the quality of the final 3D editing results by eliminating some floating discrete Gaussians, addressing fragmentation issues, and ensuring the structural continuity of the edited 3D object.

Table 3: **Human perception evaluation** for different settings. The inclusion of each module achieves more effective editing of 3D results that align with human preferences.

| ID | Settings | EC $(1 \sim 4) \uparrow$ | EA $(1 \sim 4) \uparrow$ | EQ $(1 \sim 4) \uparrow$ |
|----|----------|------|------|------|
| 0 | naive method | 1.75 | 1.625 | 1.625 |
| 1 | 0 + Primary-view Sampler | 1.875 | 1.875 | 2.125 |
| 2 | 1 + MoVE-LoRA | 2.5 | 2.625 | 2.4375 |
| 3 | 2 + Full-view Refiner | 3.875 | 3.875 | 3.8125 |

existing training-free methods, our approach necessitates more computational resources for model training. However, it achieves more precise and consistent 3D editing. The *Pro3D-Editor* framework also differs from existing methods in 2D-guided 3D editing. Existing methods typically generate a new 3D object directly from 2D multi-views without considering the structural features of the original 3D object. In contrast, our method employs the concept of sparse 3DGS reconstruction for 3D editing, which is more time-consuming than existing methods in obtaining a refined 3D structure.

## C.2 Broader Impacts

**Positive Societal Impacts.** *Pro3D-Editor* brings several contributions to the field of text-guided 3D editing. By enabling semantically accurate and spatially consistent edits across multi-views, it addresses key limitations of existing training-free approaches, which often suffer from view inconsistency and structural degradation. It has the potential to lower the barrier for creating high-quality 3D content, making it easier for designers, artists, and even non-experts to customize 3D assets using intuitive language prompts. This increased accessibility could help foster broader participation in 3D content creation and may contribute to progress in areas such as digital art, gaming, and virtual reality, where interactive and editable 3D representations are becoming increasingly important.

**Negative Societal Impacts.** Despite its advantages, the use of AI-driven 3D editing tools may also raise concerns about potential misuse. As the modification of 3D assets becomes easier and more automated, issues related to ownership, copyright infringement, and unauthorized replication of proprietary 3D models may arise. The ability to edit and redistribute high-quality 3D content with minimal expertise could blur the lines between original and derivative works, making it more challenging to protect the intellectual property rights of creators. Currently, the protection of original creators often relies on ethical norms rather than enforceable legal mechanisms, which may be insufficient to deter misuse in practice.

