# OpenReview forum: "Pro3D-Editor: A Progressive-Views Perspective for Consistent and Precise 3D Editing"
_NeurIPS.cc/2025/Conference — NeurIPS 2025 poster_

### Official Review · Reviewer_BzhZ · 2025-06-29

**Clarity:** 3
**Significance:** 2
**Originality:** 3
**Rating:** 4
**Confidence:** 4

**Summary:**

This paper proposes Pro3D-Editor, a novel progressive-views framework for text-guided 3D editing that focuses on improving spatial consistency across views. The method introduces a progressive editing strategy starting from the most salient view and propagating editing semantics to other key views using a specially designed MoVE-LoRA module, followed by a full-view refinement stage. Experiments on multiple editing prompts demonstrate clear improvements in editing quality and consistency over prior works.

**Questions:**

1. How sensitive is the final 3D editing quality to the accuracy of the Primary-view Sampler?
2. Could the authors provide a discussion on failure cases, especially when the Primary-view Sampler selects an incorrect primary view? Will it degrade to existing view-indiscriminate editing methods?
3. How does the inference time of Pro3D-Editor compare with other editing methods?
4. Is there any way to reduce the per-edit LoRA fine-tuning overhead, such as reusing or sharing LoRA?

**Ethical Concerns:**

["NO or VERY MINOR ethics concerns only"]

**Final Justification:**

While the results are promising and the method shows strong editing performance, the computational overhead remains relatively high for per-edit usage. I believe the paper has merit but would benefit from further refinement on efficiency and broader applicability, so I will keep my rating.

**Limitations:**

The progressive-view edit strategy depends on the accuracy of the Primary-view Sampler, and yet the paper does not discuss the consequences of selecting an incorrect primary view. The authors could provide a more thorough discussion on this aspect, and consider mitigation strategies accordingly.

**Quality:**

2

**Strengths And Weaknesses:**

**Strengths:**
+ The writing is clear and easy to follow.

+ The proposed progressive-view editing strategy is intuitive and experimentally shown to be effective at improving view consistency.

+ The experiments and ablations are well-designed.

**Weaknesses:**
+ The method is computationally expensive, requiring per-edit LoRA fine-tuning and a multi-stage inference pipeline.

+ The overall editing performance seems heavily depends on the Primary-view Sampler. The paper lacks a thorough discussion on its accuracy and does not explore the consequences if the selected primary view is incorrect.

+ The paper does not include inference time comparisons with other baselines. Given that Pro3D-Editor is a training-based approach, including such results is crucial for a fair and practical assessment.

+ No failure cases are discussed.

---

> ### Author Rebuttal · Authors · 2025-07-30
>
> Thank you for your valuable suggestions and constructive comments, which are crucial for improving the quality of our manuscript. Below, we address each in detail.
> # Weakness 1 & Weakness 3 & Question 3:
> The method is computationally expensive, requiring per-edit LoRA fine-tuning and a multi-stage inference pipeline. &
> The paper does not include inference time comparisons with other baselines. Given that Pro3D-Editor is a training-based approach, including such results is crucial for a fair and practical assessment.
> &
> How does the inference time of Pro3D-Editor compare with other editing methods?
> # Response 1 & 2 & 3:
> Following your advice, we provide a time cost comparison with other methods in the table below. We clarify that our method mainly focuses on improving the performance upper bound of 3D editing within practically acceptable time budgets. Our editing accuracy and consistency far surpass existing methods, achieving a 47.4% improvement in LPIPS. And for similar edits on the same 3D object (e.g., edit the same doll wearing different hats), our method can reuse the trained LoRA weights. In this case, our method requires only inference, taking roughly 10 minutes, comparable to the runtime of baselines. In contrast, other existing methods (e.g., MVEdit) cannot achieve such precise and consistent 3D object local editing.
> Besides, the time cost of our method is acceptable in light of the improvements we delivers. In real-world 3D asset production, artists often spend several hours editing a single object, much longer than ours. Our method improves editing quality while minimizing time cost. Compared with our substantial performance improvement, the time cost is practically acceptable.
> Meanwhile, in existing AIGC research, we believe that generation quality is much more important than generation efficiency. For example, fine-tuning methods like DreamBooth remain in active production use because they outperform tuning-free approaches. In the future, we will explore training a universal encoder-decoder model for multi-view editing to improve editing efficiency.
> |Method|Editing time|
> |------|------|
> |Tailor3D|4 min|
> |LGM|5 sec|
> |MVEdit|6 min|
> |3DAdapter|10 min|
> |Ours|50 min (MoVE-Lora fine-tuning) + 30s (Key-views inference) + 30min (Repair module fine-tuning) + 10min (3DGS updating)|
>
> # Weakness 2:
> The overall editing performance seems heavily depends on the Primary-view Sampler. The paper lacks a thorough discussion on its accuracy and does not explore the consequences if the selected primary view is incorrect.
> # Response 4:
> We clarify that our editing performance heavily depends on the Primary-view Sampler. Thus, we improve the accuracy of our Primary-view Sampler by leveraging CLIP similarity while simultaneously accounting for the editing information density across multiple views. The accuracy of our Primary-view Sampler is high, reaching 95%. We expand our benchmark to 40 complex cases. Results show that, in most cases, the Primary-view Sampler yields a sensible score distribution, choosing a suitable primary view. Only 2 of 40 examples select a not best primary view, leading to implausible 3D edits.
> In extreme cases, e.g., when the editing prompt is entirely irrelevant to the 3D object, no view exhibits editing saliency, causing the 2D editor to apply an unreasonable edit to the primary view, ultimately leading to 3D editing failure. Such cases are exceedingly rare. Because every view carries no editing information density in these rare cases, our method degrades into view-indiscriminate editing and the results closely resemble those shown for “Random Sampler” in Appendix Figure 3. For common 3D object editing tasks, our method handles them reliably.
>
> # Weakness 4 & Question 2:
> No failure cases are discussed. &
> Could the authors provide a discussion on failure cases, especially when the Primary-view Sampler selects an incorrect primary view? Will it degrade to existing view-indiscriminate editing methods?
> # Response 5 & 6:
> Following your advice, we will provide more failure cases in the revision. And failure cases closely resemble those shown for “Random Sampler” in Appendix Figure 3.
> When the Primary-view Sampler selects an incorrect primary view, our method degrades into view-indiscriminate editing method, but such cases are exceedingly rare. We clarify that such failures primarily occur when the editing prompt is irrelevant to the 3D object, such cases hardly ever occur in real-world 3D asset production. For common 3D object editing tasks, our method handles them reliably. Besides, other methods also fail to produce correct 3D edits in such rare cases. By comparison, our incorrect results preserve more original 3D object features, outperforming the incorrect outputs of other methods.
>
> # Question 1:
> How sensitive is the final 3D editing quality to the accuracy of the Primary-view Sampler?
> # Response 7:
> Our 3D editing quality heavily depends on the accuracy of the Primary-view Sampler. If the Primary-view Sampler selects a primary view that the 2D editor can not edit properly, our 3D edit will also fail. We first clarify that our Primary-view Sampler achieves 95% accuracy, it can handle common 3D object editing tasks reliably. Second, our pipeline demonstrates robustness to the choice of primary view. We conduct experiments in which any of the top-five ranked views is chosen as the primary view, and the resulting 3D edits are usually reasonable. However, selecting the primary view from the sixth- to tenth-ranked views usually yields implausible results.
> We provide quantitative metrics in the table below. Using any of the top-five ranked views as the primary view has minimal impact on editing quality, whereas selecting views ranked sixth to tenth leads to a noticeable drop in performance. It clearly shows that our method is robust.
> |Settings|FID &darr;|PSNR &uarr;|LPIPS &darr;|FVD &darr;|CLIP-T &uarr;|DINO-I &uarr;|
> |------|------|------|------|------|------|------|
> |Top ranked views|63.99|22.01|0.101|449.9|0.304|0.928|
> |Top-five ranked views|69.47|19.34|0.136|490.6|0.302|0.908|
> |Sixth- to tenth-ranked views|82.92|17.09|0.173|614.1|0.289|0.876|
>
> # Question 4:
> Is there any way to reduce the per-edit LoRA fine-tuning overhead, such as reusing or sharing LoRA?
> # Response 8:
> Our method allows the same MoVE-LoRA to be reused for similar edits on a given object, minimizing amortized time. For example, when we edit the same doll wearing different hats, we can reuse the same MoVE-LoRA, reducing inference time to 10 minutes. However, sharing a single MoVE-LoRA across distinct objects is infeasible, as it leads to multi-view inconsistencies and fails to preserve semantically relevant regions during editing.

---

> > ### Comment · Reviewer_BzhZ · 2025-08-04
> >
> > Thank you for the detailed responses and additional experiments. I appreciate the clarification on stage-wise runtime and the ablations on the Primary-view Sampler. While the results are promising and the method shows strong editing performance, the computational overhead remains relatively high for per-edit usage. I believe the paper has merit but would benefit from further refinement on efficiency and broader applicability, so I will keep my rating.

---

> > > ### Author Response · Authors · 2025-08-05
> > >
> > > Thank you very much for your reply. We can improve efficiency by reusing MoVE-LoRA for similar edits. Our proposed paradigm can also be naturally extended to a general training framework, and we plan to boost inference efficiency by replacing MoVE-LoRA with a universal multi-view editing model. As you suggested, we will further improve the efficiency and expand the application scope of our method in the future.

---

### Official Review · Reviewer_URcu · 2025-07-01

**Clarity:** 3
**Significance:** 2
**Originality:** 3
**Rating:** 4
**Confidence:** 4

**Summary:**

This paper introduces Pro3D-Editor, a three-stage pipeline for 3D editing: (1) selecting and editing a primary view using a proposed Primary-View Sampler; (2) editing six key views via the Key-View Render module with a Mixture-of-View-Experts Low-Rank Adaptation; and (3) propagating edits to all other views through a Full-View Refiner. The method is compared against several baselines on a custom benchmark of six objects and fifteen prompts, evaluating both editing quality and semantic accuracy.

**Questions:**

1. Why do the reported editing times differ between the main text and the supplementary materials?
2. How sensitive is Pro3D-Editor to hyperparameters such as \alpha (view-salience weight) and \lambda (refinement weight)? What value ranges were tested?

**Ethical Concerns:**

["NO or VERY MINOR ethics concerns only"]

**Final Justification:**

After reading the rebuttal and the reviews from other reviewers, and considering the discussion phase, most of my concerns have been addressed. I acknowledge that the proposed method outperforms existing approaches in terms of performance. However, I still believe that the time cost will be a significant bottleneck for its practical usage. Therefore, I will maintain my current rating.

**Limitations:**

Yes

**Quality:**

3

**Strengths And Weaknesses:**

**Strength:**
1. The paper is well written and easy to follow.
2. Visual results are impressive, showing precise, view-consistent edits.
3. A diverse set of metrics (FID, PSNR, LPIPS, FVD, CLIP-T, DINO-I, GPTEval3D) thoroughly evaluates both editing quality and accuracy.

**Weakness:**

1. Mask computation details are missing. It’s unclear how masks are generated in the two-stage pipeline and how much the two-stage process contributes without examples or descriptions of the masks. Would performance drop significantly if the two-stage step were removed?
2. Hyperparameter sensitivity is unexplored. The salience score in the Primary-View Sampler uses a naïvely designed weighting parameter \alpha, but no ablation studies show how the distribution changes with different \alpha values.
3. Small evaluation benchmark. Testing on only six objects and fifteen prompts limits conclusions about generalization to more complex or varied 3D assets.
4. High computation time. Pro3D-Editor requires about 1.5 hours per edit, compared to prior methods like MVEdit (~5 minutes), which may affect its practical use in real-world scenarios.

---

> ### Author Rebuttal · Authors · 2025-07-30
>
> We appreciate the reviewer’s valuable suggestions and will incorporate them into the manuscript. Below, we address each point in detail.
> # Weakness 1:
> Mask computation details are missing. It’s unclear how masks are generated in the two-stage pipeline and how much the two-stage process contributes without examples or descriptions of the masks. Would performance drop significantly if the two-stage step were removed?
> # Response 1:
> We clarify that the mask computation employs a standard practice rather than a core contribution of our method, so we omit its details in the main paper. To present our entire paradigm more clearly, our main paper focuses on the three other core modules and omits the mask-computation details. Following your advice, we provide the full implementation details of mask computation and will add these details in the revision.
> Given the original multi-view renders $I_c$ and the edited multi-views $I_e$ generated by MoVE-LoRA, we compute the pixel-wise difference $|I_e − I_c|$, remove isolated outliers, and take the remaining regions as the multi-view editing masks.
> The two-stage step mainly boosts multi-view consistency, yet its impact on final 3D-editing quality is secondary to that of our other modules (e.g., Primary-view Sampler, MoVE-LoRA, and Full-view Refiner). We provide an ablation study on removing this mask. Although removing the two-stage step causes a drop in quantitative metrics, the decline is far steeper when any of the other modules are removed.
> |Settings|LPIPS &darr;|CLIP-T &uarr;|DINO-I &uarr;|
> |------|------|------|------|
> |w/o two-stage step|0.104|0.303|0.908|
> |w/o Primary-view Sampler|0.106|0.297|0.911|
> |w/o MoVE-LoRA|0.103|0.303|0.892|
> |w/o Full-view Refiner|0.113|0.302|0.879|
> |Full modules|0.101|0.304|0.928|
>
> # Weakness 2:
> Hyperparameter sensitivity is unexplored. The salience score in the Primary-View Sampler uses a naïvely designed weighting parameter \alpha, but no ablation studies show how the distribution changes with different \alpha values.
> # Response 2:
> We clarify that our hyper-parameter $\alpha$ is robust. It performs well across a wide range of values. We provide corresponding ablation experiments about $\alpha$ here. We set $\alpha$  from 0 to 1 and sample it every 0.05. Experiments indicate an $\alpha$ either too large or too small yields an unreasonable peak distribution.
> - Small $\alpha$  (0 - 0.3): The chosen primary view gives little weight to back-side edit density, causing unreasonable peaks. For example, in the bottom half of Figure 3 in the main text, an incorrect peak emerges near 200°. Yet for the prompt “Girl wears a red dress”, the front view should be chosen first.
> - Large $\alpha$ (0.7 - 1.0): Too large a penalty weight shifts the peak. For example, in the upper half of Figure 3 in the main text, peak shifts away from the side toward front-back views, which are ill-suited for editing prompt “Car with a tail on the back”.
>
> The ablation experiments show that the hyper-parameter $\alpha$ performs well in the range 0.3 - 0.7, showcasing its robustness.
>
> # Weakness 3:
> Small evaluation benchmark. Testing on only six objects and fifteen prompts limits conclusions about generalization to more complex or varied 3D assets.
> # Response 3:
> We clarify that our benchmark is of the same scale as those in existing methods, for example, PrEditor3D contains only 18 objects and 40 prompts. Following your advice, we expand our benchmark. Our new beachmark includes global style edits (e.g., making an object transparent) and other editing operations (e.g., addition and deletion). And we provide quantitative metrics on this larger benchmark (main paper’s local edits benchmark with equal-sized global edits) in the table below. The results show that our method performs equally well on these tasks, demonstrating its versatility across a broader range of 3D object editing tasks.
> We also clarify that global editing is generally easier than local editing: the former only requires view-consistent changes, whereas the latter must additionally preserve every semantic-irrelevant region. Thus, in the main paper, we focus on the task of 3D object local editing and omits results for 3D global editing. We will present more editing results on complex 3D objects in the revision.
> |Methods|FID&darr;|PSNR&uarr;|LPIPS&darr;|FVD&darr;|CLIP-T&uarr;|DINO-I&uarr;|
> |-|-|-|-|-|-|-|
> |Tailor3D|123.68|13.76|0.308|987.2|0.280|0.712|
> |MVEdit|172.30|14.12|0.355|1701.1|0.298|0.747|
> |LGM|89.06|17.41|0.204|698.1|0.296|0.824|
> |**Ours**|66.09|21.26|0.106|482.9|0.301|0.919|
>
> # Weakness 4:
> High computation time. Pro3D-Editor requires about 1.5 hours per edit, compared to prior methods like MVEdit (~5 minutes), which may affect its practical use in real-world scenarios.
> # Response 4:
> We clarify that our method mainly focuses on improving the performance upper bound of 3D editing within practically acceptable time budgets. Our editing accuracy and consistency far surpass existing methods, achieving a 47.4% improvement in LPIPS. Besides, for similar edits on the same 3D object (e.g., edit the same doll wearing different hats), our method can reuse the LoRA to reduce the editing time to about 10 min.
> We clarify that the time cost is acceptable in light of the improvements our method delivers. In real-world 3D asset production, artists often spend several hours editing a single object, much longer than our method. Our method improves editing quality while minimizing time cost. Compared with our substantial performance improvement, the time cost is practically acceptable.
> Besides, existing 3D object editing methods such as MVEdit, though faster, often produce results that lose most original object features, making them unsuitable for real-world asset production.
> Meanwhile, in existing AIGC research, we believe that generation quality is much more important than generation efficiency. For example, fine-tuning methods like DreamBooth remain in active production use because they outperform tuning-free approaches. In the future, we will explore training a universal encoder-decoder model for multi-view editing to improve editing efficiency.
>
> # Question 1:
> Why do the reported editing times differ between the main text and the supplementary materials?
> # Response 5:
> We clarify that the editing time reported in the main text is same as that in the supplementary materials. We hypothesize this confusion stems from our splitting the editing time into two parts in the supplementary materials while reporting only the total in the main text. In the supplementary materials, Line 13 states that Primary-view Sampler + Key-view Render together take about 45 min, and Line 20 states that Full-view Refiner takes another 45 min, summing to about 1.5 h. And in the main text, Line 231 states that the entire editing process takes about 1.5h.
>
> # Question 2:
> How sensitive is Pro3D-Editor to hyperparameters such as \alpha (view-salience weight) and \lambda (refinement weight)? What value ranges were tested?
> # Response 6:
> We clarify that our hyper-parameter $\alpha$ and $\lambda$ are robust and they perform well across a wide range of values. The hyper-parameter $\alpha$ aims to balance the editing saliency between front and back views, and $\lambda$ aims to decoupling redundant features learned by MoVE-LoRA, ensuring consistency in the edited regions. Our method uses only two hyper-parameters, both of which yield good 3D-editing performance across a wide range of values.
> We test $\alpha$ from 0 to 1 in steps of 0.05. Experimental results demonstrate that values between 0.3 and 0.7 produce reasonable score peaks, thus we choose the midpoint 0.5.
> Likewise, we test $\lambda$ from 0 to 1 in steps of 0.1. Experimental results demonstrate that values between 0.3 and 0.6 best balance the spatial relationships learned by MoVE-LoRA and the base model’s intrinsic feature consistency. By balancing the influence of the base model and MoVE-LoRA on the edited regions, we ensure the multi-view spatial structure learned by MoVE-LoRA while disentangle its redundant features. We finally set $\lambda$ to the same midpoint 0.5.

---

> > ### Comment · Reviewer_URcu · 2025-08-05
> >
> > I appreciate the authors' detailed rebuttal. Having reviewed it alongside the other feedback, my understanding of the work has been greatly clarified. However, my primary concern regarding the method's computational overhead still holds.
> > The authors argue that the processing time is comparable to the hours a 3D artist might dedicate to an editing. This is a valid point, but it raises the question of whether the proposed method can consistently produce results of a quality that meets professional artistic standards. Additionally, the justification of computational cost by comparing it to methods like DreamBooth is less convincing in light of recent personalization works. More efficient, non-optimization-based approaches such as IP-Adapter [*1] and OmniControl [*2] have emerged, offering significantly lighter and faster alternatives without requiring model fine-tuning.
> >
> > [*1] Ye, Hu, et al. "Ip-adapter: Text compatible image prompt adapter for text-to-image diffusion models." arXiv preprint arXiv:2308.06721 (2023).
> >
> > [*2] Tan, Zhenxiong, et al. "Ominicontrol: Minimal and universal control for diffusion transformer." arXiv preprint arXiv:2411.15098 (2024).

---

> > > ### Author Response · Authors · 2025-08-06
> > >
> > > Thank you very much for your reply and valuable comments, which are very important for enhancing our contributions. Below, we respond to each of the concerns in detail.
> > >
> > > > "The authors argue that the processing time is comparable to the hours a 3D artist might dedicate to an editing. This is a valid point, but it raises the question of whether the proposed method can consistently produce results of a quality that meets professional artistic standards."
> > >
> > > We further present a comprehensive human study to strongly validate that our method consistently achieves professional artistic quality (which is not achieved by prior state-of-the-art methods). Specifically, we recruit 40 volunteers to score our method and baselines, presenting each volunteer with four examples. Each volunteer is asked to rate 3D objects according to their satisfaction level relative to real 3D assets. Each object is presented in a 360° orbital video, giving volunteers a comprehensive view of the edited 3D asset. We use a five-point scale, where 1 indicates "completely unlike a real 3D asset" and 5 indicates "virtually indistinguishable from a real professional artistic 3D asset". The results show that our method significantly outperforms existing methods, and our edited 3D assets are nearly on par with real 3D assets.
> > > |Methods|relative human satisfaction&uarr;|
> > > |-|-|
> > > |MVEdit|1.54|
> > > |LGM|3.99|
> > > |3DAdapter|2.21|
> > > |**Ours**|4.82|
> > >
> > > > "the justification of computational cost by comparing it to methods like DreamBooth is less convincing in light of recent personalization works."
> > >
> > > We clarify that our method primarily focuses on improving the upper bound of 3D object editing quality within practically affordable time (as demonstrated in our user study, we significantly raise relative human satisfaction to 96%, virtually matching that of real professional artistic 3D objects). Therefore, we compare our method with approaches like DreamBooth to **highlight that fine-tuning and non-fine-tuning methods are suited to different application scenarios**. Fine-tuning methods are preferred in quality-critical settings, where achieving the highest fidelity is essential. In contrast, non-fine-tuning methods such as IP-Adapter are better suited for scenarios where efficiency is prioritized and quality requirements are not so high.
> > > We would like to clarify that since the significant quality improvement of our method, our method is more suitable for scenarios demanding very high quality but having moderate efficiency requirements, whereas existing methods perform worse in these cases.
> > > Last but not least, **we would like to clarify that the significant improvements mainly arise from our novel progressive-view editing paradigm, as acknowledged by all reviewers**. For example, reviewer xREP notes, "The concept of a progressive-views paradigm for consistent and precise 3D editing by propagating editing semantics from salient views to sparse views has the potential to be a **seminal contribution**." We will release all our code and models, and sincerely hope that this new progressive-view paradigm can meaningfully contribute to the 3D editing community.
> > > Moreover, it is a natural next step to extend our progressive-view editing paradigm to more efficient frameworks, such as replacing MoVE-LoRA with a universal multi-view editing model, which will be the focus of our future research.
> > >
> > > |Method types|Application scenarios|
> > > |-|-|
> > > |fine-tuning methods (e.g., Dreambooth)|Demanding high generation quality but moderate efficiency requirements|
> > > |non-fine-tuning methods (e.g., IP-Adapter)|Demanding high efficiency but moderate generation quality requirements|
> > >
> > > We sincerely appreciate the reviewers’ valuable feedback once again.

---

### Official Review · Reviewer_xREP · 2025-07-02

**Clarity:** 3
**Significance:** 3
**Originality:** 3
**Rating:** 4
**Confidence:** 4

**Summary:**

This paper introduces a novel approach to text-guided 3D editing, aiming to precisely modify semantically relevant local 3D regions with applications in 3D games and film production. Unlike existing methods that edit 2D views indiscriminately and project them into 3D space, often leading to inconsistent multi-view results, the proposed Pro3D-Edit framework addresses these issues using a progressive-views paradigm, which propagates editing semantics from the editing-salient view to other editing sparse-views. Experiments demonstrate superior editing accuracy and spatial consistency compared to existing methods.

**Questions:**

1. Could the authors provide more adequate and varied qualitative results, as well as intermediate edited primary views and key views?
2. How are the individual steps of the editing process distributed during this 1.5-hour training period on a sheet of A100?

**Ethical Concerns:**

["NO or VERY MINOR ethics concerns only"]

**Final Justification:**

With the time cost breakdown and new quantitative results, most of my concerns have been addressed. If the author can provide additional qualitative results in the updated manuscript, I would like to recommend acceptance.

**Limitations:**

yes

**Quality:**

3

**Strengths And Weaknesses:**

### Strengths
1. The limitation of current approaches and the motivation for a new paradigm are clearly illustrated in Fig. 1, making the need for improvement easily understandable.
2. The concept of a progressive-views paradigm for consistent and precise 3D editing by propagating editing semantics from salient views to sparse views has the potential to be a seminal contribution.

### Weaknesses
1. The proposed approach requires the input 3D object to be represented as 3DGS and is incompatible with meshes, raising concerns about practical applicability and compatibility with recent 3D generation methods that directly output meshes.
2. The diversity of presented results is limited, raising concerns on the method's versatility.
3. Edited primary views and key views are not presented, which are important to understand how this paradigm works.

---

> ### Author Rebuttal · Authors · 2025-07-30
>
> Thank you for your constructive comments on our manuscript, which help us reorganize the our manuscript more clearly. We provide detailed responses to each point below.
> # Weakness 1:
> The proposed approach requires the input 3D object to be represented as 3DGS and is incompatible with meshes, raising concerns about practical applicability and compatibility with recent 3D generation methods that directly output meshes.
> # Response 1:
> We clarify that the core contribution of our method is to progressively and consistently project 2D editing features into 3D, which is not tied to any specific 3D representation. By modifying the Full-view Refiner, the proposed framework can be directly adapted to meshes. Our other modules Primary-view Sampler and Key-view Render achieves a 42.3 % improvement in LPIPS and a 6.8 % improvement in DINO-I (comparing ID-0 and ID-2 in Table 2 of the main paper), which is independent with 3D representation.
> Because the 3D object local editing task requires keeping semantically irrelevant regions unchanged, we adopt 3DGS instead of meshes. The iterative, differentiable nature of 3DGS naturally supports this requirement, making it better suited to this task. Besides, existing baselines also support only a single 3D representation. For example, LGM operates solely on 3DGS, whereas MVEdit is restricted to meshes.
> In future work, we will explore combining our current progressive editing paradigm with 3D generation models to achieve precise 3D object local editing directly on 3D meshes, for example, leveraging the encoder of 3D generation models, we can encode the mesh into a latent space and perform editing under the supervision of edited key views.
>
> # Weakness 2:
> The diversity of presented results is limited, raising concerns on the method's versatility.
> # Response 2:
> We clarify that our method is applicable to versatile 3D object editing tasks, including local, global and stylization edits. We focus on local editing in the main paper because it is more challenging: while global editing only requires multi-view consistency, local editing must additionally preserve all semantically irrelevant 3D regions.
> Following your advice, we conduct experiments that apply global style edits (e.g., making an object transparent) and other editing operations (e.g., addition and deletion). We provide quantitative metrics on a larger benchmark (main paper’s local edits benchmark with equal-sized global edits) in the table below. The results show that our method performs equally well on these tasks, demonstrating its versatility across a broader range of 3D object editing tasks.
> We also clarify that our new benchmark covers most common cases for 3D object editing. We will present more editing results on complex 3D objects in the revision. Besides, existing baselines employ similarly sized benchmarks. For example, PrEditor3D contains only 18 objects and 40 prompts.
> |Methods|FID&darr;|PSNR&uarr;|LPIPS&darr;|FVD&darr;|CLIP-T&uarr;|DINO-I&uarr;|
> |-|-|-|-|-|-|-|
> |Tailor3D|123.68|13.76|0.308|987.2|0.280|0.712|
> |MVEdit|172.30|14.12|0.355|1701.1|0.298|0.747|
> |LGM|89.06|17.41|0.204|698.1|0.296|0.824|
> |**Ours**|66.09|21.26|0.106|482.9|0.301|0.919|
>
> # Weakness 3 & Question 1:
> Edited primary views and key views are not presented, which are important to understand how this paradigm works. &
> Could the authors provide more adequate and varied qualitative results, as well as intermediate edited primary views and key views?
> # Response 3 & 4:
> We clarify that Figures 2 and 3 in Appendix present both the edited primary view and the key views, demonstrating the accuracy of our Primary-view Sampler and the consistency of the edited key views. Figures 2 and 3 in Appendix show the edited primary views in the left-most column and the edited key views in the remaining columns. For example, the left-most column of Figure 3 in Appendix shows that we selected an appropriate primary view and edited the chick into a cat. The remaining columns of Figure 3 in the Appendix show that, under the guidance of the edited primary view, our method successfully generates accurate and consistent key views. Following your advice, we will move these visualizations to a more obvious position in the main paper and provide more examples in the revision.
>
> # Question 2:
> How are the individual steps of the editing process distributed during this 1.5-hour training period on a sheet of A100?
> # Response 5:
> Following your advice, we report the per-step time cost in the table below. We clarify that our method aims to improve the performance upper bound of 3D editing within practically acceptable time budgets. Compared with prior work, our method raises LPIPS by 47.4%.
> We clarify that the time cost is acceptable in light of the improvements our method delivers. In real-world 3D asset production, artists often spend several hours editing a single object, much longer than our method. Our method improves editing quality while minimizing time cost. Compared with our substantial performance improvement, the time cost is practically acceptable.
> Meanwhile, in existing AIGC research, we believe that generation quality is much more important than generation efficiency. For example, fine-tuning methods like DreamBooth remain in active production use because they outperform tuning-free approaches. In the future, we will explore training a universal encoder-decoder model for multi-view editing to improve editing efficiency.
> |Step|Time cost|
> |------|------|
> |Primary-view Sampler|10 s|
> |MoVE-LoRA Fine-tuning (in Key-view Render)|50 min|
> |Edited Key-views Inferencing (in Key-view Render)|30 s|
> |Repair Module Fine-tuning (in Full-view Renfiner)|30 min|
> |3D Object Updating (in Full-view Renfiner)|10 min|

---

> > ### Comment · Reviewer_xREP · 2025-08-06
> >
> > Thanks for the rebuttal. With the time cost breakdown and new quantitative results, most of my concerns have been addressed. Please do include additional qualitative results in the updated manuscript. After reading the rebuttal and other reviewers' opinion, I would like to keep my positive rating.

---

> > > ### Author Response · Authors · 2025-08-06
> > >
> > > Thank you very much for your reply. We will add more qualitative results in our updated manuscript.

---

### Official Review · Reviewer_uWGa · 2025-07-03

**Clarity:** 3
**Significance:** 2
**Originality:** 3
**Rating:** 5
**Confidence:** 3

**Summary:**

This paper introduces a pipeline, Pro3D-Editor, which leverages Primary-view Sampler (to select and edit the most salient primary view), Key-view Render (to propagate the editing semantics to other key views) and Full-view Refiner (to jointly edite the entire 3D object) to achieve the consistent text-guided 3D editing, in a progressive-views paradigm. It tryies to address the challenges of inter-view consistency and intra-view discrimination in 3D editing task.

**Questions:**

- From Line120 to Line122, I still don't follow the reason why. Can authors have more detailed or visualized explanations?

**Ethical Concerns:**

["NO or VERY MINOR ethics concerns only"]

**Final Justification:**

My questions are all addressed. And I recommend to accept the paper.

**Limitations:**

- The pipeline is complex to do such simple object-level 3D editing tasks, thus leading to high computation resources (as authors mentioned).
- The proposed method is inherently limited to the object-level editing, where we can have 360 degree views. In the case of a 3D scene, this method falls. However, authors didn't clarify such point which is important and more challenging in "3D editing" task.
- The experiments are not convincing enough, which can be addressed if authors have more. Since there are already tons of such editing methods recently.

**Quality:**

2

**Strengths And Weaknesses:**

**Strengths**

- The paper is well-structured, the pipeline is technically well-designed and several components are described and demonstrated clearly. Overall this paper is easy to follow.
- Pro3D-Editor introduces a novel and well-motivated solution to conduct the consistent 3D editing task. The three conponents are straightforward and well-designed, where each of them makes great contributions to the entire task.
- Misture-of-View-Experts Low-Rank Adaption (MoVE-LoRA) is introduced to capture the feature correspondences from the primary view to other views, which is reasonable and easy to understand.
- The experiment of "Effectiveness of progressive-views paradigm" is reasonable to me, which demonstrates the functionality of proposed method instead of naive finetuning. And the improvements are clear to see from the quantitative numbers.
- The presentations of qualitative comparions are good.

**Weakness**

- The effectiveness of MoVE-LoRA seems vague to me, and its alation experiement is not enough. To strictly evaluate the effectiveness of MoVE-LoRA, it's better to change ID-2 to 0+Primary-view Sampler+Full-view Refiner. From the reported results in paper, the improvements from ID-1 to ID-2 is limited. I suggest to add another ablation as I mentioned above which can be compared to both ID-2 and ID-3.
- I also think the ablation ID-1 is not convencing enough. Can authors provide another ablation "3-Primary-view Sampler"? similar to the first point I listed. In the meantime, ID-2 is good to me.
- Depite the insights from the three-stage pipeline, this paper uses a fixed number of primary views. However, I believe the better one should support multiple primary views which supports to handle more complicated 3D editings, instead of simplistic objects shown in paper. As also depicted in Figure 3, there obviously are multiple views whose scores are similar and close to the highest. I hope the authors can have more discussions or even experiments about it, what if we use >1 non-overlap primary views?
- Lack of failure cases demonstrateion and the scope of the proposed solution can handle. Despite the method is straightforward, is it robust enough to handle all kinds of objects or prompts? What if the prompt is irrelevant equally to all views (which means we cannot select a good enough primary view at all)? What if most of views have high scores and how the selection of primary view in this case affect the results?
- Another point to better demonstrate the scope of the Pro3D-Editor which this paper not invovled. I hope the authors can have more experiments about the kinds of editing tasks. For example, the cases shown in paper all belong to one type: editing one single part (type or color) of the object. However, what if we edit the entire style or appearance? what if we edit only the geometry or only the appearance? I think the former one will further test the Full-view Refiner.
- Lack of the computing resources and time cost demonstrations.

I think authors should provide a clear scope the proposed solution can handle, whatever it is the best or not. Unless the authors can promise it is the best solution to the entire 3D editing task and have no failures any more.

---

> ### Author Rebuttal · Authors · 2025-07-30
>
> We sincerely appreciate the reviewer’s valuable and constructive comments, which are crucial for improving the quality of our manuscript. Below, we address each in detail.
> # Weakness 1&2:
> "The effectiveness of MoVE-LoRA seems vague to me, ...... I suggest to add another ablation as I mentioned above which can be compared to both ID-2 and ID-3." &
> "I also think the ablation ID-1 is not convencing enough. Can authors provide another ablation "3-Primary-view Sampler"? similar to the first point I listed, ......"
> # Response 1&2:
> Current metrics are insensitive to small-scale but critical multi-view inconsistencies, yet such subtle inconsistencies are easily perceived by humans. This limitation of the quantitative metrics causes improvements to appear limited. To support our claim:
> - Figures 2 and 3 in Appendix show that omitting the Primary-view Sampler leads to a noticeable mismatch between the 3D editing results and the editing prompts.
> - Figure 4 in Appendix reveals that without MoVE-LoRA, the edited key views exhibit severe multi-view inconsistency.
> - Figure 5 in Appendix demonstrates that removing the Full-view Refiner results in structural discontinuities caused by inconsistencies across full views.
>
> We believe these qualitative ablations, together with the user study presented in Appendix Table 1, more clearly show the effectiveness of each module than the quantitative metrics alone.
> Following your advice, we conduct ablation experiments showing that the Primary-view Sampler significantly improves alignment between the edited object and the prompt, and MoVE-LoRA enhances consistency across the key views.
> - Comparing ID-0 and ID-3, ID-3 achieves a 4.1% improvement in CLIP-T, indicating that the Primary-view Sampler mainly improves the similarity between results and prompts.
> - Comparing ID-1 and ID-3, ID-3 achieves a 4.0% improvement in DINO-I, indicating that MoVE-LoRA improves consistency among key views.
> - Comparing ID-2 and ID-3, ID-3 achieves a 5.6% improvement in DINO-I, indicating that Full-view Refiner improves consistency across full views. Because Full-view Refiner incorporates more views than MoVE-LoRA, the DINO-I improvement from ID-2 to ID-3 is larger than that from ID-1 to ID-3.
> |ID|Primary-view Sampler|MoVE-LoRA|Full-view Refiner|LPIPS &darr;|CLIP-T &uarr;|DINO-I &uarr;|
> |-|-|-|-|-|-|-|
> |0|&cross;|&check;|&check;|0.106|0.292|0.911|
> |1|&check;|&cross;|&check;|0.103|0.303|0.892|
> |2|&check;|&check;|&cross;|0.113|0.302|0.879|
> |3|&check;|&check;|&check;|0.101|0.304|0.928|
>
> # Weakness 3:
> Despite the insights from the three-stage pipeline, this paper uses a fixed number of primary views, ...... I hope the authors can have more discussions or even experiments about it, what if we use >1 non-overlap primary views?
> # Response 3:
> We clarify that selecting one primary view in our main paper is a specified design choice, intentionally aligned with our core motivation, i.e., ensuring consistency across all views at every step to achieve consistent 3D editing.
> First, using multiple primary views inevitably introduces conflicts among these views, as illustrated in Figure 1(b) of the main paper. Following your advice, we conduct an experiment. In this experiment, we add a tail to a chick. We find that the 90° and 270° views both yield high scores and capture non-overlapping regions. We first edit the 90° view and generate six key views, then replace the corresponding view with the edited 270° view, which resembles providing two primary views. The result exhibits a structural discontinuity at the tail, showing worse performance compared to using one primary view.
> Second, using a single primary view to generate other key views is sufficient for 3D object editing. The reason is that, as evidenced by existing 3D editing methods, six views suffice to cover all information required for editing. Thus, we design this progressive pipeline that proceeds from one primary view to six key views, capable of handling most common 3D object editing.
>
> # Weakness 4:
> Lack of failure cases demonstration and the scope of the proposed solution can handle. Despite the method is straightforward, is it robust enough to handle all kinds of objects or prompts? What if the prompt is irrelevant equally to all views (which means we cannot select a good enough primary view at all)? What if most of views have high scores and how the selection of primary view in this case affect the results?
> # Response 4:
> Following your advice, we will add more failure cases in the revision. We clarify that most observed failure cases stem from 2D editing failures, which fall outside the scope of our main motivation. For example, when we instruct the 2D editor to modify a doll’s right hand, it edits both of the hands, yielding a 3D result that mismatches the prompt.
> Our Primary-view Sampler achieves high accuracy. Expanding our benchmark to 40 cases, we find only two primary views are not best, yielding an overall accuracy of 95%.
> When the prompt is irrelevant to all views, the 2D editor cannot edit the primary view correctly, causing a mismatch between the 3D result and the prompt. We clarify that such cases fall outside our consideration, as the failure originates upstream in the 2D editing stage.
> Besides, it is rare for most views to have high scores, yet our method remains robust even in such cases. To support our claim, we deliberately choose a tree that appears identical from every 360° angle and a yellow star on its top. The star appears identical from the 0° and 180° viewpoints. Our prompt is "edits the yellow star to a red one". In this case, views within 0°-40°, 140°-220°, 320°-360° have high scores. Our method performs well regardless of which view is chosen as the primary view.
>
> # Weakness 5:
> Another point to better demonstrate the scope of the Pro3D-Editor which this paper not invovled. I hope the authors can have more experiments about the kinds of editing tasks, ...... However, what if we edit the entire style or appearance? what if we edit only the geometry or only the appearance? I think the former one will further test the Full-view Refiner.
> # Response 5:
> In the revision, we will explicitly state that our scope is focused on 3D objects, including the entire style or appearance edit. We plan to extend our paradigm to broader 3D editing tasks (e.g., 3D scene) in the future.
> Following your advice, we conduct experiments that apply global style edits (e.g., making an object transparent) and other editing operations (e.g., addition and deletion). We provide quantitative metrics on a larger benchmark (main paper’s local edits benchmark with equal-sized global edits) in the table below. The results show that our method performs equally well on these tasks, demonstrating its versatility.
> We clarify that global edits are omitted from the main paper because they are generally easier than local edits. Global edits only require view-consistent changes, whereas local edits must additionally preserve every semantic-irrelevant region. Besides, global edits are easier to Full-view Refiner, since this task amounts to reconstructing the object from multi-views. As Full-view Refiner is adapted from sparse-view reconstruction techniques, it is capable of handling global object editing tasks.
> |Methods|FID&darr;|PSNR&uarr;|LPIPS&darr;|FVD&darr;|CLIP-T&uarr;|DINO-I&uarr;|
> |-|-|-|-|-|-|-|
> |Tailor3D|123.68|13.76|0.308|987.2|0.280|0.712|
> |MVEdit|172.30|14.12|0.355|1701.1|0.298|0.747|
> |LGM|89.06|17.41|0.204|698.1|0.296|0.824|
> |**Ours**|66.09|21.26|0.106|482.9|0.301|0.919|
>
> # Weakness 6:
> Lack of the computing resources and time cost demonstrations.
> # Response 6:
> Following your advice, we provide our computing resources and time cost in the table below. For similar edits on the same 3D object (e.g., edit the same doll wearing different hats), our method can reuse the LoRA to reduce the editing time to about 10 min. We clarify that our method mainly focuses on improving the performance upper bound of 3D editing within practically acceptable time budgets. Our editing accuracy and consistency far surpass existing methods, achieving a 47.4% improvement in LPIPS.
> The time cost of our method is acceptable in light of the improvements we delivers. In real-world 3D asset production, artists often spend several hours editing a single object, much longer than ours. Our method improves editing quality while minimizing time cost. Compared with our substantial performance improvement, the time cost is practically acceptable.
> Meanwhile, in existing AIGC research, we believe that generation quality is much more important than generation efficiency. For example, fine-tuning methods like DreamBooth remain in active production use because they outperform tuning-free approaches. In the future, we will explore training a universal encoder-decoder model for multi-view editing to improve editing efficiency.
> |Method|Editing Time|GPU Memory Usage|
> |-|-|-|
> |Tailor3D|4 min|65 GB|
> |LGM|5 sec|10 GB|
> |MVEdit|6 min|13 GB|
> |3DAdapter|10 min|24 GB|
> |**Ours**|90 min|60 GB|
> |**Ours(reuse MoVE-LoRA)**|10min|20 GB|
>
> # Question 1:
> From Line120 to Line122, I still don't follow the reason why. Can authors have more detailed or visualized explanations?
> # Response 7:
> We clarify that our penalty term introduced in Line120-122 is tailored to base model’s specific view layout, ensuring edits are minimized on the missing viewpoints. Conditioned on a primary view, our base model generates six views at 0°, 45°, 90°, 180°, 270°, and 315°, which are not uniformly spaced. The 135° and 225° views are missing. Consequently, Full-view Refiner receives more views of the front viewpoints than the back, which degrades editing quality when large changes are required on the rear side. To mitigate this imbalance, this penalty term help choose a primary view that is editing salient while keeping any required edits on the rear as minimal as possible.

---

> > ### Comment · Reviewer_uWGa · 2025-08-02
> >
> > Thanks for providing more informations. My questions are addressed. I raise my score to 5.
> >
> > Please revise the paper accordingly.

---

> > > ### Author Response · Authors · 2025-08-02
> > >
> > > Thank you very much for your reply. We will further refine our manuscript in the revised version based on your advice.

---

### Note · Authors · 2025-08-14

Dear NeurIPS 2025 Reviewers, AC, SAC, and PC:

Thank you very much for your time and efforts throughout the review, rebuttal, and discussion phases.

Our paper receives **uniformly positive scores** in the initial review, with all four reviewers awarding a score of 4 (boardline accept). From the initial review:
- All reviewers highlight **the novelty of our motivation and method**, e.g., "a **novel and well-motivated** solution for consistent 3D editing" by reviewer uWGa, "... has the potential to be a **seminal contribution**" by reviewer xREP, and "The proposed progressive-view editing strategy is **intuitive**" by reviewer BzhZ.
- The recognition of our **impressive quantitative and qualitative results**, e.g., "The presentations of qualitative comparisons are good" by reviewer uWGa, "Visual results are impressive, showing precise, view-consistent edits." by reviewer URcu, "... effective at improving view consistency" by reviewer BzhZ.

During the rebuttal and discussion, **all reviewers engage actively** and we highly value their insightful comments, providing detailed responses to each concern. Specifically, the following major concerns are addressed:
- "Lack of scope analysis and failure case study": In the revision, we will clarify that we focus on 3D object editing and add more failure cases, noting that most failures stem from 2D editing, which falls outside the scope of our main motivation. For common 3D object editing, our method is robust in editing 3D objects.
- "The versatility and compatibility of our method": Our method focuses on the more challenging local editing task. But it also performs well on the 3D object global editing task. Our progressive-view editing strategy is not tied to any specific 3D representation. By simply redesigning the Full-view Refiner, it can be adapted to meshes.
- "Computation time": We aim to improve the upper bound of 3D editing performance within practical time constraints (much shorter than human 3D asset editing). Our method delivers a notable 47.4% quality boost, making it well-suited for scenarios requiring high quality with moderate efficiency.

All reviewers are satisfied with our rebuttal for addressing their concerns, and **they accordingly raise their score or keep their positive rating**.

We will revise our manuscript in accordance with the reviewers' advice. Thank you for your hard work and support.

Best regards.

---

### Decision · Program_Chairs · 2025-09-17

**Decision:**

Accept (poster)

**Comment:**

The paper receives four acceptance rating. Initially, the reviewers have concerns about some technical clarity, computational efficiency for each edit, and experimental results (e.g., ablation study). After the rebuttal, despite the suggested improvement on the editing runtime, all the reviewers are satisfied with the addressed issues in the rebuttal. The AC closely checks the paper, reviews, rebuttal, and the discussed points, and agrees with the assessment from reviewers. Therefore, the AC recommends the acceptance rating and encourages the authors to incorporate the suggestions raised by the reviewers in the final version, as well as releasing the model and code for reproducibility.